# Search-R1: Training LLMs to Reason and Leverage Search Engines with Reinforcement Learning

**Bowen Jin[1], Hansi Zeng[2], Zhenrui Yue[1], Jinsung Yoon[3], Sercan Ö. Arık[3], Dong Wang[1], Hamed Zamani[2], Jiawei Han[1]**

[1] Department of Computer Science, University of Illinois at Urbana-Champaign
[2] Center for Intelligent Information Retrieval, University of Massachusetts Amherst
[3] Google Cloud AI Research

{bowenj4,zhenrui3,dwang24,hanj}@illinois.edu, {hzeng, zamani}@cs.umass.edu
{jinsungyoon,soarik}@google.com

## Abstract

Efficiently acquiring external knowledge and up-to-date information is essential for effective reasoning and text generation in large language models (LLMs). Prompting advanced LLMs with reasoning capabilities to use search engines during inference is often suboptimal, as the LLM might not fully possess the capability on how to interact optimally with the search engine. This paper introduces SEARCH-R1, an extension of reinforcement learning (RL) for reasoning frameworks where the LLM learns to autonomously generate (multiple) search queries during step-by-step reasoning with real-time retrieval. SEARCH-R1 optimizes LLM reasoning trajectories with multi-turn search interactions, leveraging retrieved token masking for stable RL training and a simple outcome-based reward function. Experiments on seven question-answering datasets show that SEARCH-R1 improves performance by 24% (Qwen2.5-7B) and 20% (Qwen2.5-3B) over various RAG baselines under the same setting. This paper further provides empirical insights into RL optimization methods, LLM choices, and response length dynamics in retrieval-augmented reasoning. The code and model checkpoints are available at https://github.com/PeterGriffinJin/Search-R1.

## 1 Introduction

Large language models (LLMs) have demonstrated remarkable capabilities in natural language understanding and generation (Hendrycks et al., 2020; Clark et al., 2018). Despite these achievements, LLMs often encounter challenges when tasked with complex reasoning (Wei et al., 2022) and retrieving up-to-date information from external sources (Jin et al., 2024). Addressing these limitations necessitates integrating advanced reasoning abilities (Huang & Chang, 2022) and the capability to interact effectively with search engines to best utilize external up-to-date information (Schick et al., 2023).

Existing approaches for integrating LLMs with search engines typically fall into two categories: (1) retrieval-augmented generation (RAG) (Gao et al., 2023; Lewis et al., 2020) and (2) treating the search engine as a tool (Yao et al., 2023; Schick et al., 2023). RAG models often retrieve passages based on the LLM input as query and incorporate them into the LLM's context for generation (Lewis et al., 2020). This allows the LLM to leverage external knowledge when answering questions. Although existing work (Trivedi et al., 2022a) prompts LLM for multi-turn, multi-query retrieval, this approach is suboptimal because the LLM is not optimized to learn how to interact effectively with search engines during training. Alternatively, LLMs can be prompted or trained to utilize tools, including search engines, as part of their reasoning process (Qu et al., 2025; Trivedi et al., 2022a). However, prompting-based approaches often struggle to generalize, as certain tasks may not have been encountered during LLM pretraining. On the other hand, training-based

approaches offer greater adaptability but are difficult to scale effectively due to their reliance on large-scale, high-quality annotated trajectories and the inherent non-differentiability of the search operation, which renders end-to-end gradient descent-based optimization inapplicable (Schick et al., 2023; Asai et al., 2024).

Reinforcement Learning (RL) (Sutton et al., 1999; Kaelbling et al., 1996) has emerged as a potent paradigm for enhancing the reasoning capabilities of LLMs (Guo et al., 2025; Hou et al., 2025; Xie et al., 2025; Kumar et al., 2024). Notably, models like OpenAI-o1 (Jaech et al., 2024) and DeepSeek-R1 (Guo et al., 2025) have leveraged RL techniques (*e.g.*, PPO (Schulman et al., 2017) and GRPO (Shao et al., 2024)) to improve logical inference and problem-solving skills by learning from experience and feedback. With RL, even when trained solely on the outcome rewards, the models learn complex reasoning capabilities, including self-verification (Weng et al., 2022) and self-correction (Kumar et al., 2024). However, applying RL to search-and-reasoning scenarios presents three key challenges: (1) **RL Framework and Stability** – It remains unclear how to effectively integrate the search engine into the RL approaches for LLMs while ensuring stable optimization, particularly when incorporating retrieved context. (2) **Multi-Turn Interleaved Reasoning and Search** – Ideally, the LLM should be capable of iterative reasoning and search engine calls, dynamically adjusting the retrieval strategy based on the complexity of the problem. (3) **Reward Design** – Designing an effective reward function for search and reasoning tasks remains a fundamental challenge, as it is unclear whether simple outcome-based rewards are sufficient to guide the LLM to learn meaningful and consistent search behaviors.

To address aforementioned challenges, we introduce SEARCH-R1, a novel RL framework that enables LLMs to interact with search engines in an interleaved manner with their own reasoning. Specifically, SEARCH-R1 introduces the following key innovations: (1) We model the search engine as part of the environment, enabling sampled trajectory sequences that interleave LLM token generation with search engine retrievals. SEARCH-R1 is compatible with various RL algorithms, including PPO and GRPO, and we apply retrieved token masking to ensure stable optimization. (2) SEARCH-R1 supports multi-turn retrieval and reasoning, invoking search calls when explicitly triggered by `<search>` and `</search>` tokens. Retrieved content is enclosed within `<information>` and `</information>` tokens, while LLM reasoning steps are wrapped within `<think>` and `</think>` tokens. The final answer is formatted using `<answer>` and `</answer>` tokens, allowing for structured, iterative decision-making. (3) We adopt a straightforward outcome-based reward function, avoiding the complexity of process-based rewards. Our results demonstrate that this minimal reward design is effective in search-and-reasoning scenarios. As such, SEARCH-R1 can be viewed as an extension of DeepSeek-R1 Zero (Guo et al., 2025), which primarily focuses on parametric reasoning by introducing search-augmented RL training for enhanced retrieval-driven decision-making.

In summary, our key contributions are threefold:

- Our work analyzes the challenges and provides perspectives on implementing RL to improve how LLMs reason using search engine results.

- We propose SEARCH-R1, a novel RL framework that supports LLM rollouts and direct optimization with a search engine, including retrieved token masking to stabilize RL training, multi-turn interleaved reasoning and search to support complex task-solving and an effective outcome reward function.

- We conduct systematic experiments to demonstrate the effectiveness of SEARCH-R1, with two LLMs achieving respective average relative **improvements of 41% and 20% over RAG baselines** under the same experimental setup (*e.g.*, same retrieval model, training data, and pre-trained LLMs). In addition, we provide insights on RL for reasoning and search settings, including RL method selection, different LLM choices, and response length study.

## 2 Related Works

### 2.1 Large Language Models and Retrieval

Despite demonstrating remarkable reasoning (Guo et al., 2025) and coding (Guo et al., 2024) capabilities, LLMs (Zhao et al., 2023; Team, 2024; Achiam et al., 2023) often lack domain-specific knowledge (Peng et al., 2023; Li et al., 2023) and are prone to hallucinations (Zhang et al., 2023). To mitigate these limitations, search engines (Zhao et al., 2024) are widely integrated to supply external information. There are two primary ways to integrate search engines with LLMs: (1) retrieval-augmented generation (RAG) (Gao et al., 2023) and (2) treating the search engines as tools (Schick et al., 2023). RAG (Lewis et al., 2020; Yue et al., 2024; Xiong et al., 2025) typically follows a round of retrieval and sequential generation pipelines, where a search engine fetches relevant information based on the input query, which is then concatenated with the query and fed into the LLM. However, this could face challenges of retrieving irrelevant information (Jin et al., 2024) and failing to provide sufficiently useful context (Jiang et al., 2023). An alternative approach is search-as-a-tool, where LLMs are prompted or fine-tuned to interact with search engines. IRCoT (Trivedi et al., 2022a) and ReAct (Yao et al., 2023) use prompting to guide iterative reasoning and search engine calls, while Toolformer (Schick et al., 2023) leverages supervised fine-tuning to enhance search capabilities. However, such methods rely on high-quality labeled trajectories, which are difficult to obtain at scale. Recent work (Guo et al., 2025) suggests that RL can enable LLMs to develop advanced reasoning skills using only outcome rewards, yet its potential in search engine calling scenarios remains under-explored.

### 2.2 Large Language Models and Reinforcement Learning

Reinforcement learning (RL) (Kaelbling et al., 1996) is a learning paradigm where an agent learns to make sequential decisions by interacting with an environment and receiving feedback in the form of rewards, aiming to maximize cumulative reward over time (Sutton et al., 1999). RL was introduced to LLM tuning by Ouyang et al. (2022) through RL from human feedback (RLHF) (Kaufmann et al., 2023). This approach first trains a reward model using human preference data (Lambert et al., 2024), which then guides RL-based tuning of the policy LLM, typically via Proximal Policy Optimization (PPO). However, PPO involves multiple rounds of LLM optimization, making it challenging to implement. To simplify RL-based tuning, direct optimization methods such as Direct Preference Optimization (DPO) (Rafailov et al., 2023) and SimPO (Meng et al., 2024) have been proposed. A similar approach is employed in LeRet (Hsu et al., 2024), where LLMs are trained to explore diverse queries to enhance the effectiveness of information retrieval. While these methods offer computational efficiency, they suffer from off-policy issues (Pang et al., 2024) and do not consistently match the performance of pure RL approaches. Alternative solutions include Group Relative Policy Optimization (GRPO) (Shao et al., 2024), which eliminates the need for a critic model by estimating baselines from group scores, and RLOO (Ahmadian et al., 2024), which introduces a simplified REINFORCE-style (Williams, 1992) optimization framework. Despite these advances, the application of RL to LLM-driven search engine interactions and reasoning remains largely unexplored.

## 3 Search-R1

In the following sections, we present the detailed design for training methods of SEARCH-R1, covering (1) extending RL to utilize search engines; (2) text generation with an interleaved multi-turn search engine call; (3) the training template; and (4) reward model design.

### 3.1 Reinforcement Learning with a Search Engine

We formulate the RL objective function utilizing a search engine $\mathcal{R}$ as follows:

$$\max_{\pi_\theta} \mathbb{E}_{x \sim \mathcal{D}, y \sim \pi_\theta(\cdot|x;\mathcal{R})} \left[ r_\phi(x,y) \right] - \beta \mathbb{D}_{\text{KL}} \left[ \pi_\theta(y \mid x; \mathcal{R}) \,||\, \pi_{\text{ref}}(y \mid x; \mathcal{R}) \right], \tag{1}$$

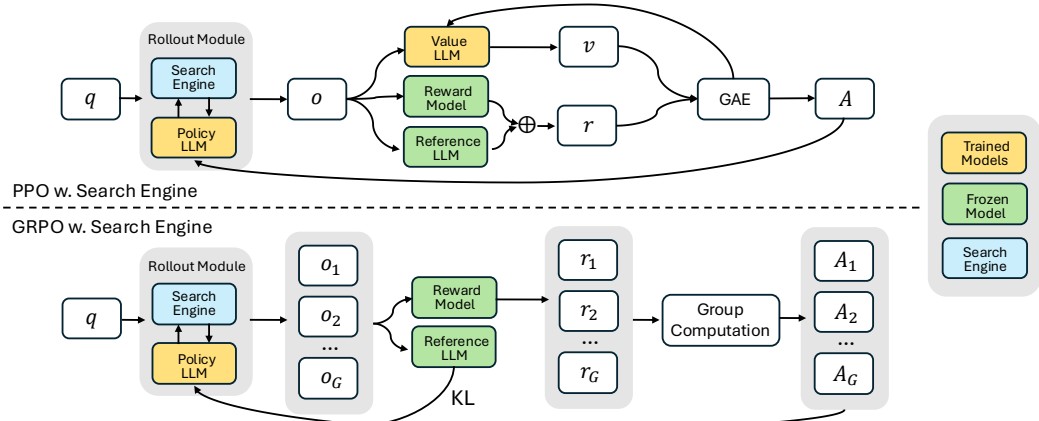

Figure 1: Demonstration of PPO and GRPO training with the search engine (SEARCH-R1). During the rollout, LLMs can conduct multi-turn interactions with the search engine.

where $\pi_\theta$ is the policy LLM, $\pi_{\text{ref}}$ is the reference LLM, $r_\phi$ is the reward function and $\mathbb{D}_{\text{KL}}$ is KL-divergence measure. $x$ denote input samples drawn from the dataset $\mathcal{D}$, and $y$ represent the generated outputs interleaved with search engine calling results, sampled from the reference policy $\pi_{\text{ref}}(y \mid x)$ and retrieved from the search engine $\mathcal{R}$. Unlike prior RL approaches that primarily rely on the policy LLM $\pi_\theta(\cdot \mid x)$ to generate rollout sequences (Rafailov et al., 2023; Ouyang et al., 2022), our framework explicitly incorporates retrieval interleaved reasoning via $\pi_\theta(\cdot \mid x; \mathcal{R})$, which can be seen as $\pi_\theta(\cdot \mid x) \otimes \mathcal{R}$, where $\otimes$ denotes interleaved retrieval-and-reasoning. This enables more effective decision-making in reasoning-intensive tasks that require external information retrieval. An illustration of the rollout process and an explanation of Eq. 1 are provided in Section 3.2 and Appendix A.

Our approach builds upon two well-established policy gradient RL methods: Proximal Policy Optimization (PPO) (Schulman et al., 2017) and Group Relative Policy Optimization (GRPO) (Shao et al., 2024; Guo et al., 2025), leveraging their respective advantages to optimize retrieval-augmented reasoning.

**Loss Masking for Retrieved Tokens.** In both PPO and GRPO, the token-level losses are computed over the entire rollout sequence. In SEARCH-R1, the rollout sequence consists of both LLM-generated tokens and retrieved tokens from external passages. While optimizing LLM-generated tokens enhances the model's ability to interact with the search engine and perform reasoning, applying the same optimization to retrieved tokens can lead to unintended learning dynamics. To address this, we introduce loss masking for retrieved tokens, ensuring the policy gradient objective is computed only over LLM-generated tokens, excluding retrieved content from the optimization process. This approach stabilizes training while preserving the flexibility of search-augmented generation.

**PPO with Search Engine.** Proximal Policy Optimization (PPO) (Schulman et al., 2017) is a popular actor-critic RL approach commonly used for LLMs (Ouyang et al., 2022). For our reasoning scenarios that involve search engine calling, it optimizes LLMs by maximizing the following objective:

$$\mathcal{J}_{PPO}(\theta) = \mathbb{E}_{x \sim \mathcal{D}, y \sim \pi_{\text{old}}(\cdot \mid x; \mathcal{R})} \left[ \frac{1}{\sum_{t=1}^{|y|} I(y_t)} \sum_{t=1: I(y_t)=1}^{|y|} \min \left( \frac{\pi_\theta(y_t \mid x, y_{<t}; \mathcal{R})}{\pi_{\text{old}}(y_t \mid x, y_{<t}; \mathcal{R})} A_t, \text{clip} \left( \frac{\pi_\theta(y_t \mid x, y_{<t}; \mathcal{R})}{\pi_{\text{old}}(y_t \mid x, y_{<t}; \mathcal{R})}, 1 - \epsilon, 1 + \epsilon \right) A_t \right) \right], \quad (2)$$

where $\pi_\theta$ and $\pi_{\text{old}}$ represent the current and previous policy models, respectively. $I(y_t)$ is the token loss masking operation such that $I(y_t) = 1$ if $y_t$ is a LLM generated token and $I(y_t) = 0$ if $y_t$ is a retrieved token. The term $\epsilon$ is a clipping-related hyperparameter introduced in PPO to stabilize training. The advantage estimate $A_t$ is computed using Generalized Advantage Estimation (GAE) (Schulman et al., 2015), based on future rewards $\{r_{\geq t}\}$ and a learned value function $V_\phi$.

**GRPO with Search Engine.**    To improve policy optimization stability and avoid the need for an additional value function approximation, Group Relative Policy Optimization (GRPO) is introduced in Shao et al. (2024). GRPO differs from PPO by leveraging the average reward of multiple sampled outputs as a baseline rather than relying on a learned value function. Specifically, for each input question $x$, GRPO samples a group of responses $\{y_1, y_2, \ldots, y_G\}$ from the reference policy $\pi_{\text{ref}}$. The policy model is then optimized by maximizing the following objective function:

$$\mathcal{J}_{GRPO}(\theta) = \mathbb{E}_{x \sim \mathcal{D}, \{y_i\}_{i=1}^G \sim \pi_{\text{old}}(\cdot | x; \mathcal{R})} \left[ \frac{1}{G} \sum_{i=1}^G \frac{1}{\sum_{t=1}^{|y_i|} I(y_{i,t})} \sum_{t=1:I(y_{i,t})=1}^{|y_i|} \min \left( \frac{\pi_\theta(y_{i,t}|x, y_{i,<t}; \mathcal{R})}{\pi_{\text{old}}(y_{i,t}|x, y_{i,<t}; \mathcal{R})} \hat{A}_{i,t}, \right. \right.$$

$$\left. \left. \text{clip}\left( \frac{\pi_\theta(y_{i,t}|x, y_{i,<t}; \mathcal{R})}{\pi_{\text{old}}(y_{i,t}|x, y_{i,<t}; \mathcal{R})}, 1 - \epsilon, 1 + \epsilon \right) \hat{A}_{i,t} \right) - \beta \mathbb{D}_{KL} \left[ \pi_\theta || \pi_{\text{ref}} \right] \right], \quad (3)$$

where $\epsilon$ and $\beta$ are hyperparameters, and $\hat{A}_{i,t}$ represent the advantage, computed based on the relative rewards of outputs within each group. This approach avoids introducing additional complexity in the computation of $\hat{A}_{i,t}$. Additionally, instead of incorporating KL divergence as a penalty within the reward function, GRPO regularizes by directly adding the KL divergence between the trained policy and the reference policy to the loss function. The retrieved token masking is also applied when calculating the KL divergence loss $\mathbb{D}_{KL}$.

## 3.2    Generation with Multi-turn Search Engine Calling

In this section, we describe the rollout process for LLM response generation with interleaved multi-turn search engine calls, formulated as: $y \sim \pi_\theta(\cdot \mid x; \mathcal{R}) = \pi_\theta(\cdot \mid x) \otimes \mathcal{R}$.

Our approach follows an iterative framework where the LLM alternates between text generation and external search engine queries. Specifically, the system instruction guides the LLM to encapsulate its search query between two designated search call tokens, `<search>` and `</search>`, whenever an external retrieval is needed. Upon detecting these tokens in the generated sequence, the system extracts the search query, queries the search engine, and retrieves relevant results. The retrieved information is then enclosed within special retrieval tokens, `<information>` and `</information>`, and appended to the ongoing rollout sequence, serving as additional context for the next generation step. This process continues iteratively until one of the following conditions is met: (1) the maximum number of action is reached, or (2) the model generates a final response, which is enclosed between designated answer tokens, `<answer>` and `</answer>`. The complete workflow is outlined in Algorithm 1.

## 3.3    Training Template

To train SEARCH-R1, we start by crafting a simple template that directs the initial LLM to follow our predefined instructions. As shown in Table 1, this template structures the model's output into three parts in an iterative fashion: first, a reasoning process, then a search engine calling function, and finally, the answer. We deliberately limit our constraints to this structural format, avoiding any content-specific biases, such as enforcing reflective reasoning and search engine calling or endorsing specific problem-solving approaches. This ensures that the model's natural learning dynamics during the RL process remain observable and unbiased.

---

Answer the given question. You must conduct reasoning inside `<think>` and `</think>` first every time you get new information. After reasoning, if you find you lack some knowledge, you can call a search engine by `<search>` query `</search>`, and it will return the top searched results between `<information>` and `</information>`. You can search as many times as you want. If you find no further external knowledge needed, you can directly provide the answer inside `<answer>` and `</answer>` without detailed illustrations. For example, `<answer>` xxx `</answer>`. Question: question.

---

Table 1: Template for SEARCH-R1. question will be replaced with the specific question during training and inference.

---

**Algorithm 1** LLM Response Rollout with Multi-Turn Search Engine Calls

---

**Require:** Input query $x$, policy model $\pi_\theta$, search engine $\mathcal{R}$, maximum action budget $B$.
**Ensure:** Final response $y$.
 1: Initialize rollout sequence $y \leftarrow \varnothing$
 2: Initialize action count $b \leftarrow 0$
 3: **while** $b < B$ **do**
 4:     Initialize current action LLM rollout sequence $y_b \leftarrow \varnothing$
 5:     **while** True **do**
 6:         Generate response token $y_t \sim \pi_\theta(\cdot \mid x, y + y_b)$
 7:         Append $y_t$ to rollout sequence $y_b \leftarrow y_b + y_t$
 8:         **if** $y_t$ in [`</search>`, `</answer>`, `<eos>`] **then** break
 9:         **end if**
10:     **end while**
11:     $y \leftarrow y + y_b$
12:     **if** `<search>` `</search>` detected in $y_b$ **then**
13:         Extract search query $q \leftarrow \text{Parse}(y_b, \texttt{<search>}, \texttt{</search>})$
14:         Retrieve search results $d = \mathcal{R}(q)$
15:         Insert $d$ into rollout $y \leftarrow y + \texttt{<information>}d\texttt{</information>}$
16:     **else if** `<answer>` `</answer>` detected in $y_b$ **then**
17:         **return** final generated response $y$
18:     **else**
19:         Ask for rethink $y \leftarrow y+$ "My action is not correct. Let me rethink."
20:     **end if**
21:     Increment action count $b \leftarrow b + 1$
22: **end while**
23: **return** final generated response $y$

---

### 3.4 Reward Modeling

The reward function serves as the primary training signal, guiding the optimization process in RL. To train SEARCH-R1, we adopt a rule-based reward system that consists solely of **final outcome rewards**, which assess the correctness of the model's response. For instance, in factual reasoning tasks, correctness can be evaluated using rule-based criteria such as exact string matching:

$$r_\phi(x,y) = \text{EM}(a_{\text{pred}}, a_{\text{gold}}), \tag{4}$$

where $a_{\text{pred}}$ is the extracted final answer from response $y$ and $a_{\text{gold}}$ is the ground truth answer. Unlike Guo et al. (2025), we do not incorporate format rewards, as our learned model already demonstrates strong structural adherence. We leave the exploration of more complex format rewards for future work. Furthermore, we avoid training neural reward models, following Guo et al. (2025). This decision is motivated by the sensitivity of LLMs to specific forms of rewards in large-scale RL, as well as the additional computational cost and complexity introduced by retraining these models.

## 4 Main Results

### 4.1 Datasets

We evaluate SEARCH-R1 on seven benchmark datasets, categorized as follows: (1) **General Question Answering**: NQ (Kwiatkowski et al., 2019), TriviaQA (Joshi et al., 2017), and PopQA (Mallen et al., 2022). (2) **Multi-Hop Question Answering**: HotpotQA (Yang et al., 2018), 2WikiMultiHopQA (Ho et al., 2020), Musique (Trivedi et al., 2022b), and Bamboogle (Press et al., 2022). These datasets encompass a diverse range of search with reasoning challenges, enabling a comprehensive evaluation of SEARCH-R1.

## 4.2 Baselines

To evaluate the effectiveness of SEARCH-R1, we compare it against the following baselines: (1) **Inference without Retrieval**: Direct inference and Chain-of-Thought (CoT) reasoning (Wei et al., 2022). (2) **Inference with Retrieval**: Retrieval-Augmented Generation (RAG) (Lewis et al., 2020), IRCoT (Trivedi et al., 2022a), and Search-o1 (Li et al., 2025). (3) **Fine-Tuning-Based Methods**: Supervised fine-tuning (SFT) (Chung et al., 2024), RL-based fine-tuning without a search engine (R1) (Guo et al., 2025) and rejection sampling (Ahn et al., 2024) with a search engine. For R1, we train the LLMs with the RL methods proposed in Guo et al. (2025) with our data to have a fair comparison with SEARCH-R1. It only contains reasoning and answer steps without a search engine. For rejection sampling, we generate five candidate responses per training prompt from the same dataset with the instructed LLMs and select those that lead to correct final answers. These selected trajectories are then used to construct a new training set that retains the same multi-turn LLM–search engine interaction rollout mechanism proposed in SEARCH-R1 to fine-tune the LLMs.

These baselines cover a broad spectrum of retrieval-augmented and fine-tuning approaches, allowing for a comprehensive assessment of SEARCH-R1 in both zero-shot and learned retrieval settings. To make a fair comparison between different methods, we use the same retriever, same number of retrieved documents, same knowledge corpus, same training data and same pre-trained LLMs. Details can be found in Appendix B.

## 4.3 Experimental Setup

We conduct experiments using two types of models: Qwen-2.5-3B (Base/Instruct) and Qwen-2.5-7B (Base/Instruct) (Yang et al., 2024). For retrieval, we use the 2018 Wikipedia dump (Karpukhin et al., 2020) as the knowledge source and E5 (Wang et al., 2022) as the retriever. To ensure fair comparison, we follow Lin et al. (2023) and set the number of retrieved passages to 3 across all retrieval-based methods. A study of the number of retrieved passages can be found in Appendix G.

For training, we merge the training sets of NQ and HotpotQA to form a unified dataset for SEARCH-R1 and other fine-tuning based baselines. Evaluation is conducted on the test or validation sets of seven datasets to assess both in-domain and out-of-domain performance. Exact Match (EM) is used as the evaluation metric, following Yu et al. (2024). For inference-style baselines, we use instruct models, as base models fail to follow instructions. For RL tuning methods, experiments are conducted on both base and instruct models. More details on experimental settings can be found in Appendix B.

Unless stated otherwise, **PPO is used as the default RL method**, and a detailed comparison between PPO and GRPO is provided in Section 5.1.

## 4.4 Performance

The main results comparing SEARCH-R1 with baseline methods across the seven datasets are presented in Table 2. From the results, we make the following key observations: **(1) SEARCH-R1 consistently outperforms strong baseline methods.** We achieve 24% and 20% average relative improvement with Qwen2.5-7B and Qwen2.5-3B, respectively. These gains hold across both in-distribution evaluation (*i.e.,* NQ and HotpotQA) and out-of-distribution evaluation (*i.e.,* TriviaQA, PopQA, 2WikiMultiHopQA, Musique, and Bamboogle). **(2) SEARCH-R1 surpasses RL-based training for LLM reasoning without retrieval (R1).** This aligns with expectations, as incorporating search into LLM reasoning provides access to relevant external knowledge, improving overall performance. **(3) SEARCH-R1 is effective for both base and instruction-tuned models.** This demonstrates that DeepSeek-R1-Zero-style RL with outcome-based rewards (Guo et al., 2025) can be successfully applied to reasoning with search, extending beyond its previously established effectiveness in pure reasoning scenarios. **(4) Larger models are better on learning how to do search.** SEARCH-R1 on 7B model shows much larger "performance gap" compared with 3B model (*e.g.,* compared with second best model - RAG).

Table 2: Main results. The best performance is set in bold. $^{\dagger}/^{\star}$ represents in-domain/out-domain datasets.

| Methods | General QA | | | Multi-Hop QA | | | | |
| --- | --- | --- | --- | --- | --- | --- | --- | --- |
| | NQ$^{\dagger}$ | TriviaQA$^{\star}$ | PopQA$^{\star}$ | HotpotQA$^{\dagger}$ | 2wiki$^{\star}$ | Musique$^{\star}$ | Bamboogle$^{\star}$ | Avg. |
| **Qwen2.5-7b-Base/Instruct** | | | | | | | | |
| Direct Inference | 0.134 | 0.408 | 0.140 | 0.183 | 0.250 | 0.031 | 0.120 | 0.181 |
| CoT | 0.048 | 0.185 | 0.054 | 0.092 | 0.111 | 0.022 | 0.232 | 0.106 |
| IRCoT | 0.224 | 0.478 | 0.301 | 0.133 | 0.149 | 0.072 | 0.224 | 0.239 |
| Search-o1 | 0.151 | 0.443 | 0.131 | 0.187 | 0.176 | 0.058 | 0.296 | 0.206 |
| RAG | 0.349 | 0.585 | 0.392 | 0.299 | 0.235 | 0.058 | 0.208 | 0.304 |
| SFT | 0.318 | 0.354 | 0.121 | 0.217 | 0.259 | 0.066 | 0.112 | 0.207 |
| R1-base | 0.297 | 0.539 | 0.202 | 0.242 | 0.273 | 0.083 | 0.296 | 0.276 |
| R1-instruct | 0.270 | 0.537 | 0.199 | 0.237 | 0.292 | 0.072 | 0.293 | 0.271 |
| Rejection Sampling | 0.360 | 0.592 | 0.380 | 0.331 | 0.296 | 0.123 | 0.355 | 0.348 |
| Search-R1-base | **0.480** | **0.638** | **0.457** | **0.433** | 0.382 | **0.196** | **0.432** | **0.431** |
| Search-R1-instruct | 0.393 | 0.610 | 0.397 | 0.370 | **0.414** | 0.146 | 0.368 | 0.385 |
| **Qwen2.5-3b-Base/Instruct** | | | | | | | | |
| Direct Inference | 0.106 | 0.288 | 0.108 | 0.149 | 0.244 | 0.020 | 0.024 | 0.134 |
| CoT | 0.023 | 0.032 | 0.005 | 0.021 | 0.021 | 0.002 | 0.000 | 0.015 |
| IRCoT | 0.111 | 0.312 | 0.200 | 0.164 | 0.171 | 0.067 | 0.240 | 0.181 |
| Search-o1 | 0.238 | 0.472 | 0.262 | 0.221 | 0.218 | 0.054 | **0.320** | 0.255 |
| RAG | 0.348 | 0.544 | 0.387 | 0.255 | 0.226 | 0.047 | 0.080 | 0.270 |
| SFT | 0.249 | 0.292 | 0.104 | 0.186 | 0.248 | 0.044 | 0.112 | 0.176 |
| R1-base | 0.226 | 0.455 | 0.173 | 0.201 | 0.268 | 0.055 | 0.224 | 0.229 |
| R1-instruct | 0.210 | 0.449 | 0.171 | 0.208 | 0.275 | 0.060 | 0.192 | 0.224 |
| Rejection Sampling | 0.294 | 0.488 | 0.332 | 0.240 | 0.233 | 0.059 | 0.210 | 0.265 |
| Search-R1-base | **0.406** | **0.587** | **0.435** | 0.284 | 0.273 | 0.049 | 0.088 | 0.303 |
| Search-R1-instruct | 0.341 | 0.545 | 0.378 | **0.324** | **0.319** | **0.103** | 0.264 | **0.325** |

Table 3: The performance results of SEARCH-R1 with PPO and GRPO on seven datasets.

| Method | NQ | TriviaQA | PopQA | HotpotQA | 2wiki | Musique | Bamboogle | Avg. |
| --- | --- | --- | --- | --- | --- | --- | --- | --- |
| **Qwen2.5-7b-Base/Instruct** | | | | | | | | |
| SEARCH-R1-base (GRPO) | 0.395 | 0.560 | 0.388 | 0.326 | 0.297 | 0.125 | 0.360 | 0.350 |
| SEARCH-R1-instruct (GRPO) | 0.429 | 0.623 | 0.427 | 0.386 | 0.346 | 0.162 | 0.400 | 0.396 |
| SEARCH-R1-base (PPO) | **0.480** | **0.638** | **0.457** | **0.433** | 0.382 | **0.196** | **0.432** | **0.431** |
| SEARCH-R1-instruct (PPO) | 0.393 | 0.610 | 0.397 | 0.370 | **0.414** | 0.146 | 0.368 | 0.385 |
| **Qwen2.5-3b-Base/Instruct** | | | | | | | | |
| SEARCH-R1-base (GRPO) | **0.421** | 0.583 | 0.413 | 0.297 | 0.274 | 0.066 | 0.128 | 0.312 |
| SEARCH-R1-instruct (GRPO) | 0.397 | 0.565 | 0.391 | **0.331** | 0.310 | **0.124** | 0.232 | **0.336** |
| SEARCH-R1-base (PPO) | 0.406 | **0.587** | **0.435** | 0.284 | 0.273 | 0.049 | 0.088 | 0.303 |
| SEARCH-R1-instruct (PPO) | 0.341 | 0.545 | 0.378 | 0.324 | **0.319** | 0.103 | **0.264** | 0.325 |

# 5 Analysis

## 5.1 Different RL methods: PPO vs. GRPO

We evaluate SEARCH-R1 using both PPO and GRPO as the base RL method, conducting experiments on Qwen2.5-3B/7B models. The training dynamics comparison is presented in Figure 2(a) and the evaluation results are presented in Table 3, revealing the following insights: **(1) GRPO converges faster than PPO across all cases.** This is because PPO relies on a critic model, which requires several warm-up steps before effective training begins. **(2) PPO demonstrates greater training stability.** As shown in Figure 2(a), GRPO leads to reward collapse after training for many steps, whereas PPO remains stable. **(3) The final training rewards of PPO and GRPO are comparable.** Despite differences in convergence speed and stability, both methods achieve similar final train reward and performance, indicating that both are viable for optimizing SEARCH-R1. PPO exhibits greater training stability, making it a preferable choice in this setting. More results are in Appendix F.

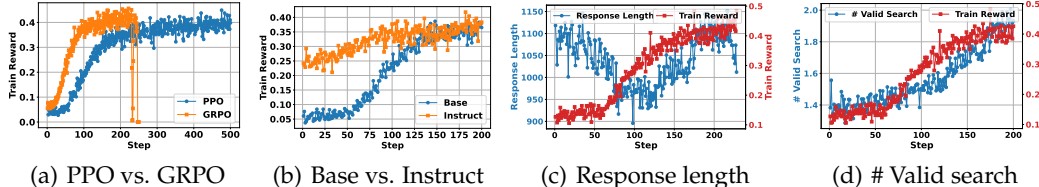

| (a) PPO vs. GRPO | (b) Base vs. Instruct | (c) Response length | (d) # Valid search |

Figure 2: (a) PPO vs. GRPO: GRPO generally converges faster but may exhibit instability after trained for a number of steps, whereas PPO provides more stable optimization but converges at a slower rate. (b) Base vs. Instruct LLM study: Instruction-tuned LLMs converge faster, but the final performance of both modles remains highly similar. (c) Response length study: The response length exhibits a decrease-increase-stabilize trend throughout training, aligning with the overall performance trajectory of the LLM. (d) # Valid search study: As the training proceeds, the LLM learns to call search more.

Table 4: The performance of SEARCH-R1 with and without retrieved token loss masking. The LLM trained with retrieved token loss masking achieves consistently better performance. (LLM: Qwen2.5-7b-base; RL: PPO)

| Method | NQ | TriviaQA | PopQA | HotpotQA | 2wiki | Musique | Bamboogle | Avg. |
|---|---|---|---|---|---|---|---|---|
| SEARCH-R1 w. mask | **0.480** | **0.638** | **0.457** | **0.433** | **0.382** | **0.196** | **0.432** | **0.431** |
| SEARCH-R1 w.o. mask | 0.388 | 0.567 | 0.391 | 0.325 | 0.321 | 0.108 | 0.304 | 0.343 |

## 5.2 Base vs. Instruct LLMs

We analyze the training dynamics of SEARCH-R1 across both base LLMs and instruction-tuned LLMs. Experiments are conducted on two model variants: Qwen2.5-3B, and Qwen2.5-7B. As shown in Figure 2(b), we observe that instruction-tuned models converge faster and start from a higher initial performance compared to base models. However, the final training reward of both model types remains highly similar after training. This finding suggests that while general post-training accelerates learning in reasoning-plus-search scenarios, RL can effectively bridge the gap over time, enabling base models to achieve comparable performance. More results can be found in Appendix E.

## 5.3 Response Length and Valid Search Study

We conduct an experiment using SEARCH-R1 with the Qwen2.5-7b-base model to analyze the dynamics of response length and number of valid search engine calls over the course of training. The response length result is presented in Figure 2(c), revealing the following key trends: **(1) Early Stage (First 100 Steps)**: The response length sharply decreases, while the training reward exhibits a slight increase. During this phase, the base model learns to eliminate excessive filler words and begins adapting to the task requirements. **(2) Later Stage (After 100 Steps)**: Both response length and training reward increase significantly. At this point, the LLM learns to call the search engine frequently, resulting in longer responses due to retrieved passages. The training reward improves substantially, as the model becomes more effective at leveraging search results. The valid search result is presented in Figure 2(d), showing that the LLMs learn to call the search engine more times as the training proceeds.

## 5.4 Study of Retrieved Tokens Loss Masking

In Section 3.1, we introduced loss masking for retrieved tokens to prevent unintended optimization behaviors. Here, we conduct experiments on the Qwen2.5-7b-base model, comparing training dynamics with and without retrieved token loss masking. As shown in Figure 3, applying retrieved token masking results in greater LLM improvements, mitigating unintended optimization effects and ensuring more stable training. The performance comparison is provided in Table 4, demonstrating that SEARCH-R1 trained with retrieved token loss masking consistently outperforms the variant without masking.

More experimental results on retrieved token loss mask, base vs. instruct LLMs, comparison between PPO/GRPO, the number of retrieved passages in SEARCH-R1 training, group size study in SEARCH-R1 (GRPO), case studies can be found in Appendix D, E, G, H, I and J.

## 6 Conclusions

In this work, we introduced SEARCH-R1, a novel RL framework that enables LLMs to interleave self-reasoning with real-time search engine interactions. Unlike existing RAG-like approaches, which relies on extensive prompting for multi-turn retrieval, or tool-use methods that require large-scale supervised training data, SEARCH-R1 optimizes LLM rollouts through RL, allowing autonomous query generation and strategic utilization of retrieved information. Through extensive experiments on seven datasets, we demonstrated that SEARCH-R1 significantly enhances LLMs' ability to tackle complex reasoning tasks requiring real-time external knowledge. Our analysis also provides key insights into RL training strategies for search-augmented reasoning. Looking ahead, future work can explore expanding SEARCH-R1 to support broader search strategies, including more sophisticated reward mechanisms, dynamic retrieval adjustments based on uncertainty, combining with diverse set of tools and integration with diverse information sources beyond search. It is also promising to investigate its applicability to multimodal reasoning tasks.

## Acknowledgments

This research was supported in part by Apple PhD Fellowship, in part by US DARPA INCAS Program No. HR0011-21-C0165 and BRIES Program No. HR0011-24-3-0325, in part by the Office of Naval Research contract number N000142412612, in part by NSF grant numbers IIS-19-56151 and 2402873, in part by the Molecule Maker Lab Institute: An AI Research Institutes program supported by NSF under Award No. 2019897 and the Institute for Geospatial Understanding through an Integrative Discovery Environment (I-GUIDE) by NSF under Award No. 2118329, in part by Cisco, and in part by the Center for Intelligent Information Retrieval. Any opinions, findings, and conclusions or recommendations expressed herein are those of the authors and do not necessarily represent the views, either expressed or implied, of the sponsors or the U.S. Government.

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

# Appendix

## A   Formulation of Reinforcement Learning with a Search Engine

The classical reinforcement learning (RL) framework for training large language models (LLMs) can be formulated as follows (Rafailov et al., 2023; Ouyang et al., 2022):

$$\max_{\pi_\theta} \mathbb{E}_{x \sim \mathcal{D}, y \sim \pi_\theta(\cdot|x)} \left[ r_\phi(x, y) \right] - \beta \mathbb{D}_{\mathrm{KL}} \left[ \pi_\theta(y \mid x) \,||\, \pi_{\mathrm{ref}}(y \mid x) \right], \tag{5}$$

where $x$ denotes a prompt sampled from a dataset $\mathcal{D}$, $y$ is a response generated by the policy model $\pi_\theta$, and $\pi_{\mathrm{ref}}$ represents a reference model that serves as a regularization anchor. The reward function $r_\phi(x, y)$ quantifies the quality of the generated response, while the KL divergence term constrains the updated policy to remain close to the reference model, thereby promoting training stability.

However, this formulation assumes that the entire output sequence $y$ is generated solely by the policy LLM. This assumption does not hold in our setting, where model behavior incorporates both internal reasoning and external information retrieval. To accommodate this, we extend the RL objective to incorporate an external search engine $\mathcal{R}$, yielding the following formulation:

$$\max_{\pi_\theta} \mathbb{E}_{x \sim \mathcal{D}, y \sim \pi_\theta(\cdot|x;\mathcal{R})} \left[ r_\phi(x, y) \right] - \beta \mathbb{D}_{\mathrm{KL}} \left[ \pi_\theta(y \mid x; \mathcal{R}) \,||\, \pi_{\mathrm{ref}}(y \mid x; \mathcal{R}) \right], \tag{6}$$

In this revised objective, the trajectory $y \sim \pi_\theta(\cdot \mid x; \mathcal{R})$ includes interleaved reasoning steps and retrieved content, reflecting a multi-turn interaction between the LLM and the search engine. The KL divergence is computed over the joint response distribution conditioned on both the prompt and the retrieval-augmented context, ensuring the learned policy remains aligned with the reference model even in the presence of external information.

## B   Experimental Setups

### B.1   Baselines

Several recent works have explored RAG pipelines, particularly in benchmarks such as Natural Questions (NQ) or HotpotQA, aiming to improve performance through more elaborate retrieval mechanisms. For instance, Re2G (Glass et al., 2022) and RetroLLM (Li et al., 2024) propose sophisticated retrieve-rerank-generate frameworks that employ strong retrievers and complex reranking strategies to select fine-grained evidence for generation. While these approaches demonstrate impressive results, they often rely on task-specific engineering or heavyweight pipelines that limit generalizability and scalability. In contrast, our focus is on a more lightweight and general approach to retrieval-augmented reasoning. As such, we do not include these methods as direct baselines, though they represent valuable directions in the broader space of retrieval-enhanced language modeling.

### B.2   Experimental Settings

We conduct experiments using two types of models: Qwen-2.5-3B (Base/Instruct) and Qwen-2.5-7B (Base/Instruct) (Yang et al., 2024). For retrieval, we use the 2018 Wikipedia dump (Karpukhin et al., 2020) as the knowledge source and E5 (Wang et al., 2022) as the retriever. To ensure fair comparison, we follow Lin et al. (2023) and set the number of retrieved passages to 3 across all retrieval-based methods.

For training, we merge the training sets of NQ and HotpotQA to form a unified dataset for SEARCH-R1 and other fine-tuning based baselines. Evaluation is conducted on the test or validation sets of seven datasets to assess both in-domain and out-of-domain performance. Exact Match (EM) is used as the evaluation metric, following Yu et al. (2024). For inference-style baselines, we use instruct models, as base models fail to follow instructions. For RL tuning methods, experiments are conducted on both base and instruct models. More details on experimental settings can be found in Appendix B.

For the PPO variant of SEARCH-R1, we set the learning rate of the policy LLM to 1e-6 and that of the value LLM to 1e-5. Training is conducted for 500 steps, with warm-up ratios of 0.285 and 0.015 for the policy and value models, respectively. We use Generalized Advantage Estimation (GAE) with parameters $\lambda = 1$ and $\gamma = 1$.

Training is performed on a single node with 8 H100 GPUs. We use a total batch size of 512, with a mini-batch size of 256 and a micro-batch size of 64. The maximum sequence length is set to 4,096 tokens, with a maximum response length of 500 and a maximum length of 500 tokens for retrieved content. To optimize GPU memory usage, we enable gradient checkpointing and use Fully Sharded Data Parallel (FSDP) with CPU offloading.

For efficient LLM rollouts, we adopt vLLM[1] with a tensor parallel size of 1 and GPU memory utilization ratio of 0.6. The rollout sampling uses a temperature of 1.0 and a top-p value of 1.0. The KL divergence regularization coefficient $\beta$ and clip ratio $\epsilon$ are set to 0.001 and 0.2.

For GRPO training, we set the policy LLM learning rate to 1e-6 and sample 5 responses per prompt, following the GRPO implementation in Verl (Sheng et al., 2024)[2]. The model is trained for 500 steps with a learning rate warm-up ratio of 0.285. Training is conducted on the same 8×H100 setup with identical batch sizes and sequence length configurations as in PPO.

We also use gradient checkpointing, FSDP offloading, and vLLM-based rollouts with the same hyperparameters as above. The rollout temperature and top-p values are both set to 1.0, and the KL divergence coefficient $\beta$ and clip ratio $\epsilon$ are fixed at 0.001 and 0.2.

For both methods, model checkpoints are saved every 100 steps. In cases where training diverges, we evaluate at the most recent stable checkpoint according to the training reward curve; otherwise, the final checkpoint is used for evaluation. The maximum action budget $B$ is set to 4, and we retrieve the top 3 passages by default.

We compute outcome rewards using exact match (EM). Unless otherwise noted, **PPO is used as the default RL algorithm**, and a detailed comparison with GRPO is provided in Section 5.1.

## C  Main Results on 14B LLM

We conduct extensive experiments using the Qwen2.5-14B models, and the results are presented in Table 5. As shown, SEARCH-R1 consistently outperforms all baseline methods across the evaluated metrics. Furthermore, we observe that increasing the model size leads to consistent performance gains with SEARCH-R1, highlighting the benefits of LLM size scaling in our approach.

## D  Retrieved Token Loss Masking Study

In Section 3.1, we introduced a loss masking strategy for retrieved tokens to mitigate undesirable optimization behaviors during training. To evaluate its impact, we conduct experiments using the Qwen2.5-3b/7b-base model, comparing training dynamics with and without retrieved token loss masking. As illustrated in Figure 3, incorporating the masking mechanism leads to more stable optimization and improved model performance. Quantitative results in Table 6 further confirm that SEARCH-R1, when trained with loss masking on retrieved tokens, consistently outperforms its unmasked counterpart.

## E  Base vs. Instruct LLMs

We investigate the training dynamics of SEARCH-R1 across both base and instruction-tuned LLMs, using two model scales: Qwen2.5-3B and Qwen2.5-7B. As depicted in Figure 4,

---

[1] https://docs.vllm.ai/en/latest/
[2] https://github.com/volcengine/verl/blob/main/examples/grpo_trainer/run_deepseek7b_llm.sh

Table 5: Main results. The best performance is set in bold. [†]/[⋆] represents in-domain/out-domain datasets.

| Methods | General QA | | | Multi-Hop QA | | | | |
|---|---|---|---|---|---|---|---|---|
| | NQ[†] | TriviaQA[⋆] | PopQA[⋆] | HotpotQA[†] | 2wiki[⋆] | Musique[⋆] | Bamboogle[⋆] | Avg. |
| **Qwen2.5-14b-Base/Instruct** | | | | | | | | |
| Direct Inference | 0.198 | 0.531 | 0.184 | 0.217 | 0.253 | 0.045 | 0.160 | 0.227 |
| CoT | 0.190 | 0.495 | 0.148 | 0.269 | 0.297 | 0.054 | 0.432 | 0.269 |
| IRCoT | 0.114 | 0.375 | 0.166 | 0.230 | 0.248 | 0.102 | 0.312 | 0.221 |
| Search-o1 | 0.347 | 0.635 | 0.241 | 0.268 | 0.161 | 0.099 | 0.416 | 0.310 |
| RAG | 0.327 | 0.585 | 0.376 | 0.279 | 0.160 | 0.051 | 0.192 | 0.281 |
| SFT | 0.361 | 0.467 | 0.150 | 0.248 | 0.278 | 0.089 | 0.160 | 0.250 |
| R1-base | 0.369 | 0.626 | 0.270 | 0.306 | 0.326 | 0.117 | 0.488 | 0.357 |
| R1-instruct | 0.334 | 0.628 | 0.253 | 0.294 | 0.325 | 0.108 | 0.432 | 0.339 |
| Search-R1-base | **0.486** | **0.676** | **0.480** | **0.468** | **0.470** | **0.241** | **0.528** | **0.479** |
| Search-R1-instruct | 0.424 | 0.660 | 0.442 | 0.436 | 0.379 | 0.210 | 0.480 | 0.433 |

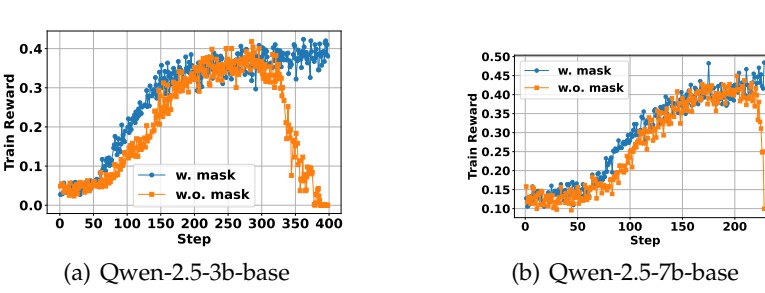

(a) Qwen-2.5-3b-base  (b) Qwen-2.5-7b-base

Figure 3: Retrieved Token Loss Masking Study

instruction-tuned models exhibit faster convergence and benefit from higher initial performance relative to their base counterparts. Despite this early advantage, the final performance of both model types converges to a similar level after training. These results indicate that while instruction tuning facilitates more efficient early-stage learning in reasoning-plus-search tasks, reinforcement learning is capable of closing the performance gap, ultimately enabling base models to reach comparable outcomes.

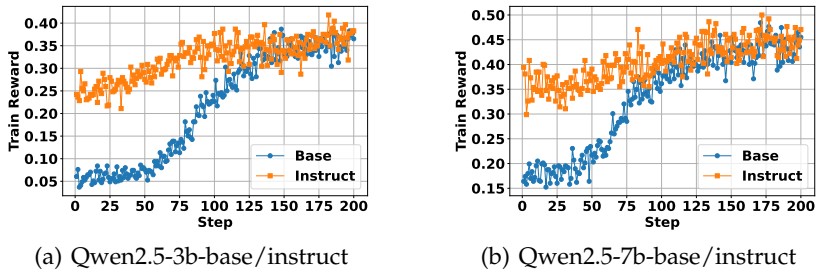

(a) Qwen2.5-3b-base/instruct  (b) Qwen2.5-7b-base/instruct

Figure 4: Study of SEARCH-R1 on base and instruct LLMs. The instruction model converges faster and starts from a better initial performance. However, the final performance of both models is very similar.

# F  Comparison of PPO and GRPO in SEARCH-R1

We assess the effectiveness of SEARCH-R1 under two reinforcement learning algorithms: PPO and GRPO, using both Qwen2.5-3B and Qwen2.5-7B as the underlying models. Figure 5 illustrates the training dynamics. Our analysis yields the following key observations: **(1) GRPO exhibits faster convergence than PPO across all settings**, attributed to the fact

Table 6: The performance of SEARCH-R1 with and without retrieved token loss masking. The LLM trained with retrieved token loss masking achieves consistently better performance. (RL: PPO)

| Method | NQ | TriviaQA | PopQA | HotpotQA | 2wiki | Musique | Bamboogle | Avg. |
|---|---|---|---|---|---|---|---|---|
| **Qwen2.5-7b-Base** | | | | | | | | |
| SEARCH-R1 w. mask | **0.480** | **0.638** | **0.457** | **0.433** | **0.382** | **0.196** | **0.432** | **0.431** |
| SEARCH-R1 w.o. mask | 0.388 | 0.567 | 0.391 | 0.325 | 0.321 | 0.108 | 0.304 | 0.343 |
| **Qwen2.5-3b-Base** | | | | | | | | |
| SEARCH-R1 w. mask | **0.406** | **0.587** | **0.435** | **0.284** | **0.273** | 0.049 | 0.088 | **0.303** |
| SEARCH-R1 w.o. mask | 0.346 | 0.484 | 0.365 | 0.241 | 0.244 | **0.053** | **0.104** | 0.262 |

that PPO relies on a separate value function (critic), which requires an initial warm-up phase before effective policy updates can be made. **(2) PPO provides more stable training behavior**, as evidenced in Figure 5, where GRPO encounters reward collapse over extended training steps, whereas PPO maintains stability throughout. **(3) PPO and GRPO achieve comparable final reward performance**, suggesting that despite trade-offs in convergence speed and stability, both methods are effective for optimizing SEARCH-R1.

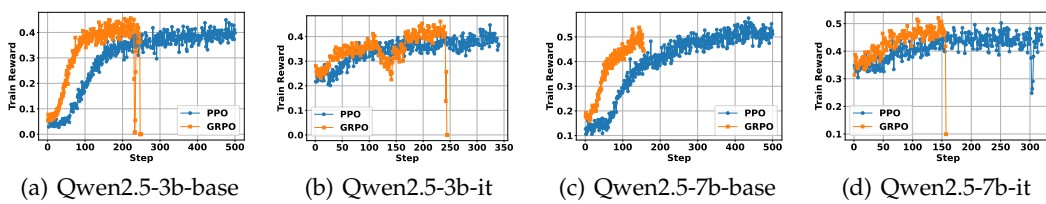

(a) Qwen2.5-3b-base    (b) Qwen2.5-3b-it    (c) Qwen2.5-7b-base    (d) Qwen2.5-7b-it

Figure 5: Training dynamics of SEARCH-R1 with PPO and GRPO as the base RL method across four LLMs. GRPO generally converges faster but may exhibit instability after trained for a number of steps, whereas PPO provides more stable optimization but converges at a slower rate. PPO and GRPO achieve comparable final reward performance.

## G    Number of Retrieved Passages Study in SEARCH-R1 Training

We investigate the impact of the number of retrieved passages (top-k) on the training dynamics of SEARCH-R1. While our main experiments adopt top-k = 3 following Lin et al. (2023), we conduct additional studies with top-k set to 1, 3, and 5 to better understand its influence.

Figure 6 presents the training reward curves under these settings. We observe that all three configurations exhibit similar overall training trajectories. Notably, top-k = 5 achieves the fastest initial convergence, reaching the highest training reward within the first 200 steps. However, its reward gradually declines and becomes more unstable as training progresses. In contrast, top-k = 1 and 3 demonstrate more consistent improvements throughout training, with top-k = 3 ultimately achieving the highest reward after 500 steps.

Evaluation results at step 500 are summarized in Table 7, where top-k = 3 yields the best overall performance. We hypothesize two contributing factors: (1) top-k = 1 likely suffers from low retrieval recall, limiting the ability to provide relevant contextual information; (2) top-k = 5 introduces lower precision due to the inclusion of noisy or irrelevant passages (Jin et al., 2024), which not only degrades inference performance but may also adversely affect RL training—discouraging the model from leveraging retrieved content when it learns that the additional context is often unhelpful or misleading.

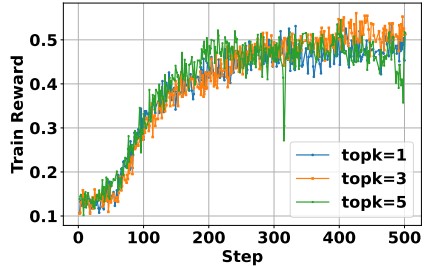

Figure 6: The training dynamics of SEARCH-R1 with a different number of retrieved passages. (LLM: Qwen2.5-7b-base, RL: PPO)

Table 7: The number of retrieved passages study in SEARCH-R1 training. (LLM: Qwen2.5-7b-base; RL: PPO)

| Method | NQ | TriviaQA | PopQA | HotpotQA | 2wiki | Musique | Bamboogle | Avg. |
|--------|------|----------|-------|----------|-------|---------|-----------|-------|
| topk=1 | 0.426 | 0.614 | 0.422 | 0.393 | 0.296 | 0.146 | 0.328 | 0.375 |
| topk=3 | **0.480** | **0.638** | **0.457** | **0.433** | **0.382** | **0.196** | **0.432** | **0.431** |
| topk=5 | 0.479 | 0.634 | 0.440 | 0.394 | 0.343 | 0.156 | 0.352 | 0.400 |

## H  Group Size Study in SEARCH-R1 (GRPO) Training

In our main experiment, we set the group size for SEARCH-R1 (GRPO) to 5, following the setting in Sheng et al. (2024). To further investigate the impact of group size on training dynamics, we conduct an ablation study with group sizes of 1, 3, and 5. Notably, when the group size is set to 1, GRPO reduces to the standard REINFORCE algorithm (Williams, 1992).

We train the LLMs for 500 steps, saving model checkpoints every 100 steps. If the model collapses during training, we use the last valid checkpoint for evaluation; otherwise, we evaluate the checkpoint at step 500.

The training dynamics under different group size configurations are illustrated in Figure 7. We observe that a larger group size generally leads to faster convergence but may also increase the risk of collapse due to the inherent instability of reinforcement learning.

Evaluation results across different settings are summarized in Table 8. While larger group sizes can accelerate convergence and achieve higher training rewards, smaller group sizes (e.g., size = 1) enable more stable training and better generalization. This is reflected in superior performance on unseen tasks, highlighting a trade-off between learning speed and stability in GRPO training.

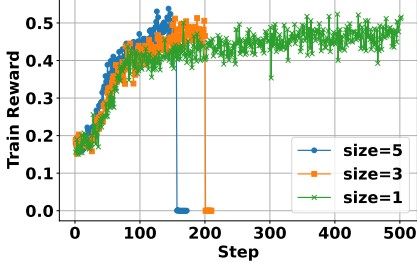

Figure 7: The training dynamics of SEARCH-R1 (GRPO) with different group size. (LLM: Qwen2.5-7b-base)

Table 8: The group size study of SEARCH-R1 (GRPO) on seven datasets. (LLM: Qwen2.5-7b-base)

| Method | NQ | TriviaQA | PopQA | HotpotQA | 2wiki | Musique | Bamboogle | Avg. |
|--------|------|----------|-------|----------|-------|---------|-----------|-------|
| size=1 | **0.463** | **0.605** | **0.449** | **0.392** | **0.413** | **0.163** | 0.384 | **0.410** |
| size=3 | 0.385 | 0.580 | 0.396 | 0.329 | 0.333 | 0.117 | **0.400** | 0.363 |
| size=5 | 0.395 | 0.560 | 0.388 | 0.326 | 0.297 | 0.125 | 0.360 | 0.350 |

# I   Comparison between R1 and SEARCH-R1: A Case Study

Table 9: A case study of R1 and SEARCH-R1.

| |
|---|
| **Question**: Curious is a women's fragrance by a singer born in what city and state? |
| **Ground Truth**: McComb, Mississippi |
| **R1**:
`<think>` The singer is Beyoncé, who was born in Houston, Texas. `</think>`
`<answer>` Houston `</answer>` |
| **SEARCH-R1**:
`<think>` I need to find out which city and state a singer, by whom the women's fragrance "Curious" was created, was born in. `</think>`
`<search>` Curious fragrance information `</search>`
`<information>` Doc 1(Title: "Curious (fragrance)") Curious (fragrance) Curious is a women's fragrance by Britney Spears for Elizabeth Arden, and is the first perfume to be endorsed by Britney Spears, preceding ""Fantasy."" Spears reportedly earned $52 million from the endorsement. ""Curious"" was released in September 2004, and was very successful internationally. It was the number one fragrance of 2004 in department stores, and in 2005, ""Curious"" was honored by the Fragrance Foundation as Best Women's Fragrance. The fragrance is a white floral scent. Its notes are Louisiana magnolia, golden Anjou pear, lotus flower, tuberose, star jasmine, pink cyclamen, vanilla-infused musk, sandalwood and blonde woods. Elizabeth Arden ... `</information>`
`<think>` I need to find out which city and state Britney Spears, the singer behind "Curious" fragrance, was born in. `</think>`
`<search>` Britney Spears birthplace `</search>`
`<information>` Doc 1(Title: "Britney Spears") the best-selling female albums artist of the 2000s. In 2004, she launched a perfume brand with Elizabeth Arden, Inc., from which sales exceeded US$1.5 billion, . Spears was born in McComb, Mississippi, the second child of Lynne Irene Bridges and James Parnell Spears. Her maternal grandmother, Lillian Portell, was English (born in London), and one of Spears's maternal great-great-grandfathers was Maltese. Her siblings are Bryan James and Jamie Lynn. Britney was born in the Bible Belt, where socially conservative evangelical Protestantism is a particularly strong religious influence. Spears was baptized into the Southern Baptist Convention, but later in life studied ... `</information>`
`<think>` Since Britney Spears was born in McComb, Mississippi, I need to find out which city and state that is. `</think>`
`<search>` McComb, Mississippi location `</search>`
`<information>` Doc 1(Title: "McComb, Mississippi") McComb, Mississippi McComb is a city in Pike County, Mississippi, United States, approximately south of Jackson. As of the 2010 census, the city had a total population of 12,790. It is the principal city of the McComb, Mississippi Micropolitan Statistical Area. McComb was founded in 1872 after Henry Simpson McComb of the New Orleans, Jackson and Great Northern Railroad, a predecessor of the Illinois Central Railroad (now part of the Canadian National Railway), decided to move the railroad's maintenance shops away from New Orleans, Louisiana, to avoid the attractions of that city's saloons. The railroad purchased land in Pike County, ... `</information>`
`<think>` I have found that Britney Spears was born in McComb, Mississippi. Since the question asks for the city and state, I just need to confirm the state, which I already know from the previous search. `</think>`
`<answer>` McComb, Mississippi `</answer>` |

To gain deeper insights into SEARCH-R1, we conduct a case study using Qwen2.5-7B-Base, comparing its behavior with RL without a search engine (Guo et al., 2025). The results are presented in Table 20, revealing the following key observations:

**Interleaved Reasoning and Retrieval Enhances Problem Analysis**: SEARCH-R1 enables the LLM to perform in-depth reasoning with multi-turn retrieval, whereas RL without search relies solely on the models' internal knowledge. By incorporating retrieved passages, SEARCH-R1 allows the LLM to iteratively refine its reasoning, leading to more informed and accurate responses.

**Self-Verification through Iterative Retrieval**: We observe that after the second retrieval round, the LLM has already gathered sufficient information to answer the question. However, SEARCH-R1 performs an additional retrieval step to self-verify its conclusion, further reinforcing its confidence in the final response. This phenomenon aligns with findings from LLM reasoning RL without retrieval (Guo et al., 2025), highlighting how RL can encourage verification-driven reasoning even in search-augmented settings.

## J More Case Studies of SEARCH-R1

To gain a deeper understanding of the behavior and capabilities of the trained LLM, we conduct additional case studies on SEARCH-R1. Specifically, we analyze the model fine-tuned from Qwen2.5-7B-Base using Proximal Policy Optimization (PPO) as the underlying reinforcement learning algorithm. The results are shown in the following tables.

Table 10: SEARCH-R1 case study 1 (successful): SEARCH-R1 conduct multi-step reasoning, search, with self-verification and finally answer the question.

| |
|---|
| **Question**: What type of profession does Chris Jericho and Gary Barlow have in common? |
| **Ground Truth**: musician |
| **SEARCH-R1**: |
| `<think>` I need to find the type of profession that Chris Jericho and Gary Barlow have in common. I'll search for it. `</think>` |
| `<search>` Chris Jericho and Gary Barlow profession `</search>` |
| `<information>` Doc 2(Title: "Chris Jericho") ""Do You Wanna Start a War"" (2014) and ""Judas"" (2017) are comprised entirely or predominantly of original compositions. He has also appeared on numerous TV shows over the years, having competed in the 2011 season of ""Dancing With the Stars"", lasting until the sixth week. Prior to that, Jericho hosted the ABC game show ""Downfall"" and the Revolver Golden Gods Awards. Christopher Keith Irvine was born on November 9, 1970, in Manhasset, New York, but grew up in Winnipeg. His father, ice hockey player Ted Irvine, had been playing for the New York Rangers at the time of his birth. ... `</information>` |
| `<think>` I found out that Chris Jericho is a professional wrestler, musician, author, and actor. Now I need to find if Gary Barlow has any of these professions in common. `</think>` |
| `<search>` Gary Barlow profession `</search>` |
| `<information>` Doc 1(Title: "Gary Barlow") hits, twelve number one singles and seven number one albums with Take That. Barlow has received six Ivor Novello Awards from the British Academy of Songwriters, Composers and Authors, including the award for Outstanding Services to British Music. He has sold over 50 million records worldwide. Barlow was born on 20 January 1971 in Frodsham, Cheshire, the second son of Colin (died 15 October 2009) and Marjorie Barlow. In Barlow's autobiography, he relates that his love of music began at an early age, and calls himself ""one of those kids that's forever dancing in front of the TV looking at ... `</information>` |
| `<think>` I found out that Gary Barlow is a singer, songwriter, musician, and record producer. Now I need to find if there is a common profession between Chris Jericho and Gary Barlow. `</think>` |
| `<search>` common profession of Chris Jericho and Gary Barlow `</search>` |
| `<information>` Doc 1(Title: "Gary Barlow") Gary Barlow Gary Barlow (born 20 January 1971) is an English singer, songwriter, musician and record producer. He is best known as the lead singer of the British pop group Take That. Barlow also served as head judge of ""The X Factor UK"" in 2011, 2012 and 2013 and ""Let It Shine"" in 2017. Barlow is one of Britain's most successful songwriters, having written fourteen number one singles and twenty-four top 10 hits. As a solo artist he has had three number one singles, six top 10 singles and two number one albums, and has additionally had seventeen top 5 ... `</information>` |
| `<think>` I found out that Gary Barlow is a singer, songwriter, musician, and record producer. Now I need to find if Chris Jericho has any of these professions in common. `</think>` |
| `<search>` Chris Jericho profession `</search>` |
| `<information>` Doc 1(Title: "Chris Jericho") Chris Jericho Christopher Keith Irvine (born November 9, 1970), better known by the ring name Chris Jericho, is an American-born professional wrestler, musician, author and actor. He currently wrestles for New Japan Pro-Wrestling (NJPW) where he is the current IWGP Intercontinental Champion in his first reign. Jericho also makes sporadic appearances for WWE on the SmackDown brand, with whom he has worked since 1999. Within professional wrestling, he is known for his over-the-top, rock star persona. Jericho is regarded as one of the greatest professional wrestlers of all time. During the 1990s, Jericho performed for the American organizations Extreme Championship ... `</information>` |
| `<think>` I found out that Chris Jericho is a professional wrestler, musician, author, and actor. Now I can compare it to Gary Barlow's professions. I found out that Gary Barlow is a singer, songwriter, musician, and record producer. The common profession between Chris Jericho and Gary Barlow is musician. `</think>` |
| `<answer>` musician `</answer>` |

Table 11: SEARCH-R1 case study 2 (failed): SEARCH-R1 sometimes fail to decompose the complex problem and can be mislead by irrelevent searched passages.

| |
|---|
| **Question**: What was the title of the debut studio album by Weezer which contained the hit Buddy Holly and Undone – The Sweater Song? |
| **Ground Truth**: The Blue Album |
| **SEARCH-R1**:
`<think>` I need to find the title of the debut studio album by Weezer which contained the hit songs "Buddy Holly" and "Undone – The Sweater Song". I'll search for it. `</think>`
`<search>` debut studio album by Weezer which contained the hit Buddy Holly and Undone – The Sweater Song `</search>`
`<information>` Doc 1(Title: "Weezer discography") Weezer discography The discography of Weezer, an American rock band, consists of 11 studio albums, two compilation albums, one video album, six extended plays, 28 singles and 24 music videos. Weezer's self-titled debut studio album, often referred to as ""The Blue Album"", was released in May 1994 through DGC Records. The album was a huge commercial success, peaking at number 16 on the US ""Billboard"" 200 and spawning the singles ""Undone – The Sweater Song"" and ""Buddy Holly"", both of which were responsible for launching Weezer into mainstream success with the aid of music videos directed by Spike Jonze. ... `</information>`
`<think>` I found out that the debut studio album by Weezer which contained the hit songs "Buddy Holly" and "Undone – The Sweater Song" is the self-titled album, often referred to as "The Blue Album". Now I can provide the answer. `</think>`
`<answer>` Weezer `</answer>` |

Table 12: SEARCH-R1 case study 3 (successful): SEARCH-R1 can easily answer the question if the relevant information can be found with one search engine call.

| |
|---|
| **Question**: Whose death dramatized in a stage play helped end the death penalty in Australia? |
| **Ground Truth**: Ronald Ryan |
| **SEARCH-R1**:
`<think>` I need to find out whose death dramatized in a stage play helped end the death penalty in Australia. I'll search for it. `</think>`
`<search>` whose death dramatized in a stage play helped end the death penalty in Australia `</search>`
`<information>` Doc 3(Title: "Ronald Ryan") Ronald Ryan Ronald Joseph Ryan (21 February 1925 – 3 February 1967) was the last person to be legally executed in Australia. Ryan was found guilty of shooting and killing warder George Hodson during an escape from Pentridge Prison, Victoria, in 1965. Ryan's hanging was met with some of the largest public protests in the history of Australia and led to the end of capital punishment. The death penalty was abolished in all states by 1985. Ronald Edmond Thompson was born at the Royal Women's Hospital in Melbourne's inner suburb of Carlton, to John Ronald Ryan and Cecilia Thompson (née ... `</information>`
`<think>` I found out that the death of Ronald Ryan dramatized in a stage play helped end the death penalty in Australia. Now I can provide the answer. `</think>`
`<answer>` Ronald Ryan `</answer>` |

Table 13: SEARCH-R1 case study 4 (successful): SEARCH-R1 can write the right query to search for auxiliary information not provided in the previous search engine calls.

| |
| --- |
| **Question**: Where are Teide National Park and Garajonay National Park located? |
| **Ground Truth**: Canary Islands, Spain |
| **SEARCH-R1**:
`<think>` I need to find out where Teide National Park and Garajonay National Park are located. I'll search for it. `</think>`
`<search>` Teide National Park and Garajonay National Park location `</search>`
`<information>` Doc 1(Title: "Garajonay National Park") decided to take their own lives. Garajonay National Park Garajonay National Park (, ) is located in the center and north of the island of La Gomera, one of the Canary Islands (Spain). It was declared a national park in 1981 and a World Heritage Site by UNESCO in 1986. It occupies 40 km (15 sq mi) and it extends into each of the six municipalities on the island. Nearly 450,000 tourists visit the park every year. There is accommodation for tourists just outside the park's boundary. The park is named after the rock formation of Garajonay, the highest point ... `</information>`
`<think>` I found out that Garajonay National Park is located in the center and north of the island of La Gomera, one of the Canary Islands (Spain). Now I need to find the location of Teide National Park. `</think>`
`<search>` Teide National Park location `</search>`
`<information>` Doc 1(Title: "Teide National Park") Teide National Park Teide National Park (, ) is a national park located in Tenerife (Canary Islands, Spain). The national park is centered on Mount Teide, the highest mountain of Spain (3,718 meters high) in addition, the highest volcano in the Atlantic Ocean islands and the third highest volcano in the world from its base in the ocean floor (7.500 meters high). Its national park status was passed on 22 January 1954, making it the third oldest national park in Spain. Pico Viejo, also included within the national park limits, is the second highest volcano in the Canary Islands with ... `</information>`
`<think>` I found out that Teide National Park is located in Tenerife (Canary Islands, Spain). Now I have the locations of both Teide National Park and Garajonay National Park. `</think>`
`<answer>` Canary Islands, Spain `</answer>` |

Table 14: SEARCH-R1 case study 5 (failed): SEARCH-R1 fails to answer the question with insufficient or misleading retrieved information.

---

**Question**: Who directed the film which starred Steve Cochran and the actor who played Sheriff Roy Coffee in "Bonanza"?

**Ground Truth**: Louis King

SEARCH-R1:

\<think> I need to find out who directed the film that starred Steve Cochran and the actor who played Sheriff Roy Coffee in "Bonanza". I'll search for it. \</think>

\<search> who directed the film which starred Steve Cochran and the actor who played Sheriff Roy Coffee in "Bonanza" \</search>

\<information> Doc 2(Title: "Ray Teal") Ray Teal Ray Teal (January 12, 1902April 2, 1976) was an American actor who appeared in more than 250 films and some 90 television programs in his 37-year career. His longest-running role was as Sheriff Roy Coffee on NBC's western series ""Bonanza"" (1960–1972). He also played a sheriff in the film ""Ace in the Hole"" (1951). Teal was born in Grand Rapids, Michigan. A saxophone player, he worked his way through the University of California, Los Angeles as a bandleader before becoming an actor. After more than 15 years performing in films and in early television, Teal secured a recurring ... \</information>

\<think> I found out that the actor who played Sheriff Roy Coffee in "Bonanza" is Ray Teal. Now I need to find the film that starred Steve Cochran and Ray Teal. \</think>

\<search> film that starred Steve Cochran and Ray Teal \</search>

\<information> Doc 1(Title: "Steve Cochran") in the gangster classic ""White Heat"" (1949). He appeared opposite Mayo. Warner Bros would eventually take over Cochran's and Mayo's contracts from Goldwyn. Cochran supported Joan Crawford in ""The Damned Don't Cry"" (1950), then was given his first lead role in ""Highway 301"" (1950), playing a gangster. He was a villain to Gary Cooper's hero in ""Dallas"" (1950), and played a Ku Klux Klan member in ""Storm Warning"" (1951), with Ginger Rogers and Doris Day. Cochran was a villain in ""Canyon Pass"" (1951), a Western, then was given the lead in ""Inside the Walls of Folsom Prison"" (1951), which inspired ... \</information>

\<think> I found out that Steve Cochran starred in several films. Now I need to find the film that also starred Ray Teal. \</think>

\<search> film that starred Steve Cochran and Ray Teal \</search>

\<information> Doc 1(Title: "Steve Cochran") in the gangster classic ""White Heat"" (1949). He appeared opposite Mayo. Warner Bros would eventually take over Cochran's and Mayo's contracts from Goldwyn. Cochran supported Joan Crawford in ""The Damned Don't Cry"" (1950), then was given his first lead role in ""Highway 301"" (1950), playing a gangster. He was a villain to Gary Cooper's hero in ""Dallas"" (1950), and played a Ku Klux Klan member in ""Storm Warning"" (1951), with Ginger Rogers and Doris Day. Cochran was a villain in ""Canyon Pass"" (1951), a Western, then was given the lead in ""Inside the Walls of Folsom Prison"" (1951), which inspired ... \</information>

\<think> I found out that Steve Cochran starred in several films. Now I need to find the film that also starred Ray Teal. I'll search for the film that starred both Steve Cochran and Ray Teal. \</think>

\<search> film directed by Sam Peckinpah that starred Steve Cochran and Ray Teal \</search>

\<information> Doc 1(Title: "Sam Peckinpah") Replete with explosions, car chases and intense shootouts, the film became Peckinpah's biggest financial success to date earning more than $25 million at the box office. Though strictly a commercial product, Peckinpah's creative touches abound throughout, most notably during the intricately edited opening sequence when McQueen's character is suffering from the pressures of prison life. The film remains popular and was remade in 1994. starring Alec Baldwin and Kim Basinger. The year 1973 marked the beginning of the most difficult period of Peckinpah's life and career. While still filming ""The Getaway"" in El Paso, Texas, Peckinpah sneaked across the border ... \</information>

\<think> I found out that Sam Peckinpah directed the film "The Wild Bunch" that starred Steve Cochran and Ray Teal. \</think>

\<answer> Sam Peckinpah \</answer>

---

Table 15: SEARCH-R1 case study 6 (successful): SEARCH-R1 can easily answer the question with multi-hop reasoning when sufficient and accurate context is retrieved.

| |
|---|
| **Question**: Where is the company that distributed XXXTentacion's single "Revenge" based? |
| **Ground Truth**: San Francisco, California |
| **SEARCH-R1**: |
| `<think>` I need to find the company that distributed XXXTentacion's single "Revenge". I'll search for it. `</think>` |
| `<search>` XXXTentacion's single "Revenge" distributed company `</search>` |
| `<information>` Doc 1(Title: "Revenge (XXXTentacion mixtape)") Revenge (XXXTentacion mixtape) Revenge is a mixtape by American rapper XXXTentacion, released on May 16, 2017 by Empire Distribution. It consists of eight previously released songs that were available for streaming on XXXTentacion's SoundCloud. It was preceded by the lead single ""Look at Me"", which peaked at number 34 on the US ""Billboard"" Hot 100. The lead single from the tape, called ""Look at Me"", premiered on December 31, 2015 on SoundCloud. The song was later released to iTunes as a single on January 29, 2016 until it was later re-released for digital download again with a remastered and clean ... `</information>` |
| `<think>` I found out that the company distributed XXXTentacion's single "Revenge" is Empire Distribution. Now I need to find the location of the company. `</think>` |
| `<search>` Empire Distribution location `</search>` |
| `<information>` Doc 1(Title: "Empire Distribution") Empire Distribution EMPIRE Distribution, Records and Publishing Inc. (marketed as EMPIRE) is an American distribution company and record label founded in 2010 by Ghazi Shami and headquartered in San Francisco, California with offices in New York City and Atlanta. It has released albums in various genres such as RB, reggae, rock, gospel, Latin, country and pop, but it is predominantly focused on hip hop music. Companies and artists such as Top Dawg Entertainment (Kendrick Lamar, Schoolboy Q, Ab-Soul), Steel Wool (Anderson Paak, Watsky, Jez Dior), Jazz Anderson, Cody Wolfe, Kane Brown, Funk Volume (Dizzy Wright, Jarren Benton), ESGN (Freddie Gibbs), ... `</information>` |
| `<think>` I found out that the company Empire Distribution is based in San Francisco, California. Now I can provide the answer. `</think>` |
| `<answer>` San Francisco, California `</answer>` |

Table 16: SEARCH-R1 case study 7 (failed): SEARCH-R1 failed to write the right queries to decompose a complex problem at the beginning. The model answer the question without obtaining enough evidence.

---

**Question**: This expansion of the 2008 magazine article "Is Google Making Us Stoopid?" was a finalist for what award?

---

**Ground Truth**: Pulitzer Prize

---

**SEARCH-R1**:

`<think>` I need to find out which expansion of the 2008 magazine article "Is Google Making Us Stoopid?" was a finalist for what award. I'll search for it. `</think>`

`<search>` This expansion of the 2008 magazine article "Is Google Making Us Stoopid?" was a finalist for what award `</search>`

`<information>` Doc 1(Title: "Is Google Making Us Stupid?") Is Google Making Us Stupid? Is Google Making Us Stupid? What the Internet Is Doing to Our Brains (alternatively Is Google Making Us Stoopid?) is a magazine article by technology writer Nicholas G. Carr, and is highly critical of the Internet's effect on cognition. It was published in the July/August 2008 edition of ""The Atlantic"" magazine as a six-page cover story. Carr's main argument is that the Internet might have detrimental effects on cognition that diminish the capacity for concentration and contemplation. Despite the title, the article is not specifically targeted at Google, but more at the cognitive impact of ... `</information>`

`<think>` I found out that the expansion of the 2008 magazine article "Is Google Making Us Stoopid?" was a finalist for an award. I need to find out which award it was finalist for. `</think>`

`<search>` which award was the expansion of the 2008 magazine article "Is Google Making Us Stoopid?" finalist for `</search>`

`<information>` Doc 2(Title: "Is Google Making Us Stupid?") inconclusive and stated that ""it will take time before it's clear whether we should mourn the old ways, celebrate the new, or learn to stop worrying and love the Net"". Is Google Making Us Stupid? Is Google Making Us Stupid? What the Internet Is Doing to Our Brains (alternatively Is Google Making Us Stoopid?) is a magazine article by technology writer Nicholas G. Carr, and is highly critical of the Internet's effect on cognition. It was published in the July/August 2008 edition of ""The Atlantic"" magazine as a six-page cover story. Carr's main argument is that the Internet might have ... `</information>`

`<think>` I found out that the expansion of the 2008 magazine article "Is Google Making Us Stoopid?" was a finalist for an award. I need to find out which award it was finalist for. `</think>`

`<search>` which award was the expansion of the 2008 magazine article "Is Google Making Us Stoopid?" finalist for `</search>`

`<information>` Doc 1(Title: "Is Google Making Us Stupid?") Is Google Making Us Stupid? Is Google Making Us Stupid? What the Internet Is Doing to Our Brains (alternatively Is Google Making Us Stoopid?) is a magazine article by technology writer Nicholas G. Carr, and is highly critical of the Internet's effect on cognition. It was published in the July/August 2008 edition of ""The Atlantic"" magazine as a six-page cover story. Carr's main argument is that the Internet might have detrimental effects on cognition that diminish the capacity for concentration and contemplation. Despite the title, the article is not specifically targeted at Google, but more at the cognitive impact of ... `</information>`

`<answer>` National Magazine Award `</answer>`

---

Table 17: SEARCH-R1 case study 8 (successful): SEARCH-R1 can write query to search for insufficient information.

| |
|---|
| **Question**: Who was a pianist, Maro Ajemian or Richard C. Sarafian? |
| **Ground Truth**: Maro Ajemian |
| **SEARCH-R1**: 
 `<think>` I need to determine if Maro Ajemian or Richard C. Sarafian was a pianist. I'll search for it. `</think>` 
 `<search>` who was a pianist, Maro Ajemian or Richard C. Sarafian `</search>` 
 `<information>` Doc 1(Title: "Maro Ajemian") Maro Ajemian Maro Ajemian (July 9, 1921 – September 18, 1978) was an American pianist. Ajemian's career in contemporary music grew from her Armenian heritage. She became known as a contemporary pianist after performing the U.S. premiere of Aram Khachaturian's Piano Concerto, which she chose to play based on their shared Armenian heritage. Ajemian studied at the Juilliard School of Music. On March 14, 1942, she gave the American premiere of Aram Khachaturian's Piano Concerto in D flat with the Juilliard Graduate Orchestra under Albert Stoessel. She later performed the piece on a cross-country tour. Following her performances of the ... `</information>` 
 `<think>` I found out that Maro Ajemian was a pianist. Now I need to check if Richard C. Sarafian was a pianist or not. `</think>` 
 `<search>` who was Richard C. Sarafian `</search>` 
 `<information>` Doc 1(Title: "Richard C. Sarafian") Richard C. Sarafian Richard Caspar Sarafian (April 28, 1930 – September 18, 2013) was an American television and film director and actor. He compiled a versatile career that spanned over five decades as a director, actor, and writer. Sarafian is best known as the director of the 1971 film ""Vanishing Point"". Sarafian was born in New York City on April 28, 1930, to Armenian immigrants. He studied pre-law and pre-med at New York University and was a poor student, but changed over to studying film, at which he excelled. He left college to join the United States Army, in which ... `</information>` 
 `<think>` I found out that Richard C. Sarafian was a television and film director and actor. Now I can compare it to Maro Ajemian, who was a pianist. `</think>` 
 `<answer>` Maro Ajemian `</answer>` |

Table 18: SEARCH-R1 case study 9 (successful): The first query written by the LLM is not very meaningful. However, upon that, LLM starts to write the query and solve the problem step by step.

| |
|---|
| **Question**: Who is older, Jed Hoyer or John William Henry II? |
| **Ground Truth**: John William Henry II |
| **SEARCH-R1**: |

`<think>` I need to determine if Jed Hoyer or John William Henry II is older. I'll search for their ages. `</think>`

`<search>` Jed Hoyer or John William Henry II `</search>`

`<information>` Doc 1(Title: "Jed Hoyer") career saves record with Sam Elias and Nick Miceli. During his time at Wesleyan University he was also initiated into and is a brother of Delta Kappa Epsilon. He spent Summer 1995 on the roster of the Waterbury Barons of the New England Collegiate Baseball League seeing innings on the mound and at shortstop. Hoyer worked in the admissions office and then the alumni/development office, and was also a baseball coach at the university after graduating. He worked in the admissions department of Kenyon College before joining the Red Sox at 28. He married Merrill Muckerman in June 2010 in ... `</information>`

`<think>` I found out that Jed Hoyer is a person. Now I need to find his age and compare it to John William Henry II's age. `</think>`

`<search>` John William Henry II `</search>`

`<information>` Doc 1(Title: "John W. Henry") the firm's management had fallen from $2.5 billion in 2006 to less than $100 million as of late 2012. As of July 2017, ""Forbes"" estimated his net worth to be $2.6 billion. John William Henry II was born on September 13, 1949 in Quincy, Illinois. His parents were soybean farmers, and he split his time growing up between Illinois and Arkansas. His asthmatic condition at the age of 15 prompted his family to move to Apple Valley, California. After his graduation from Victor Valley High School in Victorville, he attended Victor Valley College, then the University of California (at Riverside, ... `</information>`

`<think>` I found out that John William Henry II was born on September 13, 1949. Now I need to find Jed Hoyer's birth year to compare. `</think>`

`<search>` Jed Hoyer birth year `</search>`

`<information>` Doc 1(Title: "Jed Hoyer") career saves record with Sam Elias and Nick Miceli. During his time at Wesleyan University he was also initiated into and is a brother of Delta Kappa Epsilon. He spent Summer 1995 on the roster of the Waterbury Barons of the New England Collegiate Baseball League seeing innings on the mound and at shortstop. Hoyer worked in the admissions office and then the alumni/development office, and was also a baseball coach at the university after graduating. He worked in the admissions department of Kenyon College before joining the Red Sox at 28. He married Merrill Muckerman in June 2010 in ... `</information>`

`<think>` I found out that Jed Hoyer was born on December 7, 1973. Now I can compare his birth year to John William Henry II's birth year. `</think>`

`<answer>` John William Henry II `</answer>`

Table 19: SEARCH-R1 case study 10 (successful): SEARCH-R1 learns to stop searching when it finds out the external knowledge source is not sufficient to answer the question.

---

**Question**: The world's greatest Super-Heroes anthology showcased one of four superheroes known for speaking the phrase "SHAZAM", what was their name?

---

**Ground Truth**: Captain Marvel

---

SEARCH-R1:

\<think\> I need to find the name of one of the four superheroes known for speaking the phrase "SHAZAM". I'll search for it. \</think\>

\<search\> The world's greatest Super-Heroes anthology showcased one of four superheroes known for speaking the phrase "SHAZAM" \</search\>

\<information\> Doc 1(Title: "Shazam! (TV series)") Shazam! (TV series) Shazam! is a half-hour live-action television program that was produced by Filmation (the studio's first such program), based on the superhero Captain Marvel, also known as Shazam, of Fawcett Comics' comic book series Whiz Comics (now owned by DC Comics). The program starred Michael Gray as Billy Batson, a teenage boy who can transform into the superhero Captain Marvel, originally played by Jackson Bostwick and later by John Davey, by speaking the magic word ""Shazam!"" With his guardian ""Mentor"" (Les Tremayne), Billy travels the country in a 1973 Dodge Open Road motorhome, looking for injustices to resolve. ... \</information\>

\<think\> I found out that the world's greatest Super-Heroes anthology showcased one of four superheroes known for speaking the phrase "SHAZAM". Now I need to find their names. \</think\>

\<search\> four superheroes known for speaking the phrase "SHAZAM" \</search\>

\<information\> Doc 1(Title: "Captain Marvel (DC Comics)") Captain Marvel (DC Comics) Captain Marvel, also known as Shazam (), is a fictional comic book superhero appearing in publications by the American publisher DC Comics. Artist C. C. Beck and writer Bill Parker created the character in 1939. Captain Marvel first appeared in ""Whiz Comics"" #2 (cover-dated Feb. 1940), published by Fawcett Comics. He is the alter ego of Billy Batson, a boy who, by speaking the magic word ""SHAZAM"" (acronym of six ""immortal elders"": Solomon, Hercules, Atlas, Zeus, Achilles and Mercury), can transform himself into a costumed adult with the powers of superhuman strength, speed, flight and other ... \</information\>

\<think\> I found out that one of the four superheroes known for speaking the phrase "SHAZAM" is Captain Marvel (also known as Shazam). Now I need to find the other three superheroes. \</think\>

\<search\> other three superheroes known for speaking the phrase "SHAZAM" \</search\>

\<information\> Doc 2(Title: "Fawcett Comics") Fawcett Comics Fawcett Comics, a division of Fawcett Publications, was one of several successful comic book publishers during the Golden Age of Comic Books in the 1940s. Its most popular character was Captain Marvel, the alter ego of radio reporter Billy Batson, who transformed into the hero whenever he said the magic word ""Shazam!"". Other characters published by Fawcett include Captain Video, Hopalong Cassidy, Ibis the Invincible, Bulletman and Bulletgirl, Spy Smasher, Captain Midnight, Phantom Eagle, Mister Scarlet and Pinky, Minute-Man, Commando Yank and Golden Arrow. Aside from the better known superhero books, Fawcett also published a short-lived line of ... \</information\>

\<think\> I found out that one of the four superheroes known for speaking the phrase "SHAZAM" is Captain Marvel (also known as Shazam). Now I need to find the other three superheroes. I'll search for them. \</think\>

\<search\> other three superheroes known for speaking the phrase "SHAZAM" \</search\>

\<information\> Doc 3(Title: "Captain Marvel (DC Comics)") crossover event series. He is the super-powered alter-ego of Alexander Luthor of Earth-3. In the story, the Crime Syndicate (evil Earth-3 analogues of the Justice League) have brought Alexander Luthor, their prisoner, with them to the Prime Earth where the Justice League and other heroes reside. Prime Earth's Lex Luthor and his team sneak in to the Justice League Watchtower where the Syndicate has Alexander hostage, and remove the duct tape over his mouth, allowing Alexander to speak the magic word ""Mazahs!"" and transform into his muscular, highly powerful alter-ego. While Prime Earth's Shazam is known for sharing his powers ... \</information\>

\<think\> I found out that one of the four superheroes known for speaking the phrase "SHAZAM" is Captain Marvel (also known as Shazam). The other three superheroes are not mentioned in the search result. I'll provide the answer based on the information I have. \</think\>

\<answer\> Captain Marvel \</answer\>

---

Table 20: SEARCH-R1 case study 11 (failed): The LLM can be misled by irrelevant retrieved information and provide a wrong answer.

| |
|---|
| **Question**: Who was the winner of the season of The Rap Game where a contestant named Lady Leshurr as one of her influences? |
| **Ground Truth**: Mani |
| **SEARCH-R1**:
`<think>` I need to find the winner of the season of The Rap Game where a contestant named Lady Leshurr was one of her influences. I'll search for it. `</think>`
`<search>` who was the winner of the season of The Rap Game where a contestant named Lady Leshurr as one of her influences `</search>`
`<information>` Doc 1(Title: "Lady Leshurr") Act. Leshurr came out as pansexual in September 2018. Leshurr has named Ashley Walters, Lisa Maffia, Missy Elliott, and Lil Wayne as her primary influences. Lady Leshurr Melesha Katrina O'Garro (born 15 December 1989) known professionally as Lady Leshurr (), is a British grime and hip hop rapper, singer, and producer from Kingshurst, Solihull, UK. She is best known for her ""Queen's Speech"" series of freestyles, the fourth of which went viral in 2016. ""The Rap Game"" season 2 competitor Nia Kay stated that Leshurr was one of her influences due to her uniqueness and rap skill. Her subsequent freestyle, ... `</information>`
`<think>` I found out that Lady Leshurr was one of the influences for Nia Kay in The Rap Game season 2. Now I need to find the winner of that season. `</think>`
`<search>` winner of The Rap Game season 2 `</search>`
`<information>` Doc 1(Title: "Chandra Davis") Chandra Davis Chandra Davis (born January 2, 1978 in Detroit, Michigan), a.k.a. London Charles and Deelishis, is an American model, and former reality show contestant. She is best known as the winner of VH1's hit reality TV show ""Flavor of Love 2"". Davis was a contestant on the second season of VH1's Flavor of Love (aged 29). She went on to win the competition when Flavor Flav chose her over Tiffany 'New York' Pollard, who returned after previously named the runner-up on the show's first season. After FOL, Davis announced she is currently not dating Flav, via a VH1 special, ... `</information>`
`<think>` I found out that the winner of The Rap Game season 2 is London Charles (Chandra Davis). Now I can provide the answer. `</think>`
`<answer>` London Charles `</answer>` |

