# OpenReview forum: "Search-R1: Training LLMs to Reason and Leverage Search Engines with Reinforcement Learning"
_colmweb.org/COLM/2025/Conference — COLM 2025_

### Official Review · Reviewer_cUvm · 2025-05-07

**Rating:** 6
**Confidence:** 4
**Ethics Flag:** 1

**Summary:**

In this paper, the authors have proposed a novel training pipeline by employing reinforcement learning directly to RAG system by treating the answer correctness as the outcome reward. It demonstrates some technical contributions and serves as one of the earliest work exploring RL with LLMs beyond mathematical reasoning. Yet, this work is highly inspired by DeepSeek-R1 such that fails to carefully discuss the necessity of employing reinforcement learning without any RAG-like SFT.

**Reasons To Accept:**

1. Serving one of the earliest work exploring RL with LLMs in retrieval-based QA, this paper demonstrates some technical contribution and provide some experience to the community.
2. The writing is easy to follow and experimental results are good.

**Reasons To Reject:**

I think the paper is highly affected by DeepSeek-R1 such that if fails to address some more important aspects, including but may not be limited to
1. Is it really necessary to perform reinforcement learning directly without any cold-start SFT? Note that the instruct-version model may also not be trained on data invoked with search engine. How will these two fashions perform differently?
2. Following point 1, the baseline that using either rejection-sampling or knowledge distillation for trajectory collection and using SFT or DPO training is missing.

---

> ### Author Response · Authors · 2025-06-01
>
> We sincerely thank the reviewer for the constructive feedback, which has strengthened our work. Please find our point-by-point responses below:
>
>
> - **Is it necessary to perform RL without any cold-start SFT data?** We agree with the reviewer that incorporating an intermediate cold-start supervised fine-tuning (SFT) stage could potentially improve final performance. However, SFT relies on large-scale, high-quality annotated trajectories of search-and-reasoning interactions, which are costly and difficult to obtain—posing significant challenges to scalability. In this work, we show that such intermediate trajectories can instead be acquired automatically through outcome-only reinforcement learning. The resulting RL-trained model can then be used to generate synthetic data for future cold-start SFT. We leave the systematic exploration of this direction to future work.
>
>
> - **Instruct-version model may also not be trained on data invoked with a search engine. How do these two fashions perform differently?** You are right. The general instructed LLMs are typically not trained on interleaved reasoning and search engine invocation data, and their performance can indeed be improved with more targeted supervision. In this work, we demonstrate that both base and instructed LLMs can learn to perform interleaved reasoning and search behavior through outcome-driven reinforcement learning. Moreover, the resulting RL-trained models can be used to generate high-quality trajectories with explicit search engine usage, which can serve as synthetic data for future supervised fine-tuning.
>
>
> - **Baseline which performs rejection sampling or knowledge distillation for trajectory collection is missing.** In this work, our primary objective is to propose a reinforcement learning (RL)-based method for training large language model (LLM) agents capable of interleaving reasoning and search. While knowledge distillation from a larger teacher model is a possible direction, it introduces additional supervision signals that may lead to unfair comparisons. That said, we agree with the reviewer that rejection sampling offers a reasonable and relevant baseline for comparison. Accordingly, we have included this baseline using both Qwen2.5-3B-it and Qwen2.5-7B-it. Specifically, we generate five candidate responses per training prompt from the Search-R1 dataset and select those that lead to correct final answers. These selected trajectories are then used to construct a new training set that retains the same multi-turn LLM–search engine interaction rollout mechanism proposed in Search-R1. We refer to this variant as Search-R1 (Rejection Sampling). The updated results are presented below, showing that Search-R1 with RL consistently outperforms Search-R1 with rejection sampling across both model sizes.
>
>
> Qwen2.5-7b-it
>
> | Method | NQ | TriviaQA | PopQA | HotpotQA | 2wikimultihopQA | Musique | Bamboogle | Avg |
> |--|--|--|--|--|--|--|--|--|
> | Direct | 0.1343 | 0.4075 | 0.1402 | 0.1831 | 0.2502 | 0.0314 | 0.1200 | 0.1810 |
> | CoT | 0.0481 | 0.1851 | 0.0539 | 0.0915 | 0.1106 | 0.0215 | 0.2320 | 0.1061 |
> | IRCoT | 0.2240 | 0.4775 | 0.3009 | 0.1331 | 0.1486 | 0.0715 | 0.2240 | 0.2257 |
> | Search-o1 | 0.1507 | 0.4429 | 0.1307 | 0.1873 | 0.1757 | 0.0583 | 0.2960 | 0.2059 |
> | RAG | 0.3490 | 0.5847 | 0.3924 | 0.2990 | 0.2348 | 0.0579 | 0.2080 | 0.3037 |
> | SFT | 0.3183 | 0.3538 | 0.1208 | 0.2173 | 0.2586 | 0.0662 | 0.1120 | 0.2067 |
> | RL w.o. search | 0.2700 | 0.5370 | 0.1990 | 0.2370 | 0.2920 | 0.0720 | 0.2930 | 0.2714 |
> | Search-R1 (Rejection Sampling) | 0.3604 | 0.5922 | 0.3797 | 0.3310 | 0.2958 | 0.1233 | 0.3548 | 0.3482 |
> | Search-R1 (RL) | **0.3925** | **0.6103** | **0.3965** | **0.3700** | **0.4142** | **0.1456** | **0.3680** | **0.3853** |
>
>
>
> Qwen2.5-3b-it
>
> | Method | NQ | TriviaQA | PopQA | HotpotQA | 2wikimultihopQA | Musique | Bamboogle | Avg |
> |--|--|--|--|--|--|--|--|--|
> | Direct | 0.1058 | 0.2879 | 0.1075 | 0.1491 | 0.2442 | 0.0199 | 0.0240 | 0.1341 |
> | CoT | 0.0227 | 0.0324 | 0.0045 | 0.0213 | 0.0208 | 0.0024 | 0.0000 | 0.0149 |
> | IRCoT | 0.1110 | 0.3117 | 0.2002 | 0.1636 | 0.1713 | 0.0666 | 0.2400 | 0.1806 |
> | Search-o1 | 0.2382 | 0.4723 | 0.2617 | 0.2211 | 0.2180 | 0.0538 | **0.3200** | 0.2550 |
> | RAG | **0.3485** | 0.5441 | **0.3866** | 0.2551 | 0.2256 | 0.0472 | 0.0800 | 0.2696 |
> | SFT | 0.2490 | 0.2923 | 0.1036 | 0.1857 | 0.2478 | 0.0443 | 0.1120 | 0.1764 |
> | RL w.o. search | 0.2100 | 0.4490 | 0.1710 | 0.2080 | 0.2750 | 0.0600 | 0.1920 | 0.2236 |
> | Search-R1 (Rejection Sampling) | 0.2942 | 0.4879 | 0.3324 | 0.2396 | 0.2327 | 0.0588 | 0.2097 | 0.2650 |
> | Search-R1 (RL) | 0.3410 | **0.5451** | 0.3784 | **0.3244** | **0.3193** | **0.1027** | 0.2640 | **0.3250** |

---

> > ### Comment · Reviewer_cUvm · 2025-06-06
> > **Thanks for the updated results**
> >
> > Thanks for the authors' efforts for clarification and more results. I think the newly updated results are strong.

---

> > > ### Author Response · Authors · 2025-06-06
> > >
> > > Thank you very much for your thoughtful feedback and for recognizing the strength of the newly updated results! We're glad to hear that our clarifications addressed your concerns effectively. We truly appreciate your consideration and would be grateful if you could consider reflecting this in your final score. We are happy to continue the discussion if you have any other questions!

---

> > > ### Author Response · Authors · 2025-06-09
> > >
> > > Hi Reviewer cUvm,
> > >
> > > As we approach the discussion deadline, we wanted to sincerely thank you again for your thoughtful feedback. We're grateful that you found the newly updated results strong, and we hope this addresses your concerns effectively.
> > >
> > > If possible, we would deeply appreciate it if **your more positive view could be reflected in your final score**. Of course, we remain happy to discuss any remaining questions you might have before the deadline.
> > >
> > > Thank you again for your time and consideration!

---

### Official Review · Reviewer_yiKE · 2025-05-12

**Rating:** 7
**Confidence:** 4
**Ethics Flag:** 1

**Summary:**

This paper proposes to incorporate retrieved tokens from external search engines in the rollout stage of reinforcement learning (PPO and GRPO). It also introduces a "loss masking" strategy to exclude retrieved tokens from model optimization, aiming to stabilize the RL training process.

- **Overall Quality**: 4 / 5
- **Clarity**: 3.5 / 5
- **Originality**: 3.5 / 5
- **Significance**: 3.5 / 5

**Questions To Authors:**

1. About "Retrieved Tokens Loss Masking":
    - Q1.1: To clarify, the retrieved tokens ("<information>d</information>") are masked out during training, while all of them actively serve as the input of the policy model $\pi_{\theta}$ during the rollout stage (as in Algorithm 1 Line 6), right?
    - Q1.2: Although the loss masking strategy stabilizes the training (Section 5.4), it encourages the model to learn to answer the question without any retrieved information (which is masked/skipped). Thus, the training process does not enhance the information retrieval ability of the model if Loss Masking is applied, right?
2. Equation 2 and Equation 3: It looks better to replace "$\sum_{t=1:I(y_t)=1}^{|y|} \min(...)$" with "$\sum_{t=1}^{|y|} I(y_t) \min(...)$"
3. Section 3 (Algorithm 1):
    - Q3.1: According to the algorithm, the rollout sequence may end up without a final answer (wrapped by <answer> </answer>). What is the ratio of such sequences, and how does this case affect the training results?
    - Q3.2: What does the Parse function exactly do? At least it will remove "<search>" and "</search>" tokens in $y_b$ and extract tokens between them, right?
    - Q3.3: Where is the prompt in Table 1 placed? Is it at the beginning of $x$ in Algorithm 1? Will the input query $x$ change during the rollout stage of the current question?
4. About the evaluation:
    - Q4.1: During inference, does Search-R1 still work like the rollout stage, where the model generates search queries and calls search engines before answering?
    - Q4.2: Some datasets in the experiment have provided context for searching. Hence, it seems unnecessary to call external search engines (as in Search-R1) to solve those tasks.
5. Will the rollout data, training code, and trained model be open-source?

**Reasons To Accept:**

1. The proposed method is novel and well-elaborated.
2. The paper is well-written, despite formatting issues.
3. This paper provides a new approach, other than RAG and tool-using, to utilize searching for reasoning.
4. The experimental results on multiple QA benchmarks demonstrate the effectiveness of the proposed method.

**Reasons To Reject:**

1. The implementation is preliminary. E.g., the simple training template and the rule-based reward model.
2. Some questions and concerns to clarify. Please refer to Questions To Authors.
3. Format issues of the paper.
    - The paper style (`\usepackage[xxx]`) is not "submission" but "preprint" or "final". Thus, there are no line numbers to refer to.
    - Table 2/3/4: The captions should be below the tables.

---

> ### Author Response · Authors · 2025-06-01
>
> We appreciate your insightful feedback and believe it has significantly strengthened our manuscript. We have carefully addressed each of your comments as detailed below:
>
> - **The implementation is preliminary.** Given that reinforcement learning for interleaved reasoning and search in LLM agents remains underexplored, we intentionally begin with a clean and simple setting to establish a strong foundation. It is encouraging to observe that Search-R1 proves to be a simple yet effective RL methodology in this context. We view this as a promising initial point, and future works can explore more complex instructions and advanced reward designs (e.g., neural reward models).
>
> - **Format.** We are using the “submission” option, but sorry, not enabling “\ifcolmsubmission”. Will fix it accordingly.
>
> - **Q1.1.** Yes, they serve as inputs for policy rollout.
>
> - **Q1.2.** To clarify, when computing the token-level loss, we mask out the retrieved tokens and only include the LLM-generated tokens in the loss calculation. However, the logits for these generated tokens are still conditioned on the retrieved tokens, as the retrieved content is provided as input during optimization. This setup ensures that while the model leverages retrieved information to inform its generation, it is explicitly trained to write effective queries and perform reasoning over the retrieved content, rather than memorizing external information.
>
> - **Q2.** Thank you for the comments. We will make modifications accordingly.
>
> - **Q3.1.** We conducted a study on Search-R1 using both Qwen2.5-3B and Qwen2.5-7B. The ratio of such sequences is 1.32% and 4.98%, respectively. Notably, this ratio is influenced by the maximum action budget B: as B increases, the ratio tends to decrease. A larger B also improves final performance, but at the cost of efficiency, as it leads to longer rollouts.
>
> - **Q3.2.** You are right. It will remove the <search> </search> tokens and extract the query in between.
>
> - **Q3.3.** Yes, the prompt in Table 1 is placed at the beginning of x. The input query x will not be changed during rollout.
>
> - **Q4.1.** Yes, Search-R1 conducts interleaved reasoning and search engine calling during inference.
>
> - **Q4.2.** In our experiments, we focus on the challenging open-domain setting where only the question is provided, without any additional context. This requires the LLMs to actively retrieve relevant information on their own. We will clarify this setup in the revised manuscript for better transparency.
>
> - **Q5.** Sure, all the resources will be open-sourced.

---

> > ### Comment · Reviewer_yiKE · 2025-06-04
> > **Response acknowledgment**
> >
> > Thanks for the reply. This work looks solid to me. Please include the mentioned points in the revision.

---

> > > ### Author Response · Authors · 2025-06-04
> > >
> > > Thank you for your feedback! We appreciate your positive assessment and will incorporate the mentioned points into the revised version.

---

### Official Review · Reviewer_udKF · 2025-05-12

**Rating:** 7
**Confidence:** 3
**Ethics Flag:** 1

**Summary:**

The paper trains LLMs with reinforcement learning (RL) to interleave chain-of-thought reasoning and search. It prompts the model to use special tokens to enclose reasoning, search, and final answer sections. Search results are provided to the model as an additional section, and loss during the RL phase is masked on search results as these cannot be influenced by the model. RL uses either proximal policy optimization (PPO) or group relative policy optimization (GRPO), with a simple exact-match reward function on the result. The setup is evaluated on 7 question-answering (QA) datasets with two Qwen-2.5 model variants (3B and 7B) in base and instruct versions. The result shows clear improvements over RAG-only and reasoning-only baselines.

**Questions To Authors:**

- In Algorithm 1, why do you explicitly force the model to decode "my action is not correct" -- you can't know that at this point, right?
- Any intuition on how this will work with larger models (30B, 70B) and models of a different family?

Minor comments:
- Fig. 1 deserves a more detailed commentary and should be referenced from the text (I didn't find any reference)
- The sentence "An illustration of the rollout proces..." (pg. 4) is unclear, did you mean to add "respectively" at the end?
- The text in Sect. 3.3 starts by implying a single-turn process, whereas the prompt clearly shows it's multi-turn – please rephrase.
- The sentence "For R1, we train..." on pg. 6 is unclear.
- I know it's because you get better results with the 7B model, but it feels like base vs. instruct is not so prominent to warrant a space in Table 2. It's also confusing because we don't know which baseline is "base" and which is "instruct".
- You say in Sect. 5.3 the training rewards gets a decrease in the first 100 steps. The chart looks more like it stagnates or grows slightly.
- What's a "valid search" in Fig. 2? Does it just mean the number of times the model produces <search></search> tokens, or do you check if the search gets any results?
- When you refer to Fig. 3 in Sect. 5.4, you should mention it's in the appendix (and perhaps stress the link to Table 4 more).

**Reasons To Accept:**

- Very reasonable overall setup with positive results
- Experiments look mostly sound, with multiple strong baselines
- Mostly clear writing

**Reasons To Reject:**

- The evaluation is relatively limited (two small variants of a single LLM, just the QA task)
- The paper claims the training is stable, but doesn't actually show results for multiple runs
- The training of the base vs. instruct variant seems a little inconsistent (base wins for 7B but instruct wins for 3B, the paper doesn't comment on this)
- I personally don't like the name grab of R1, the model is not based on DeepSeek and the experiment is only QA

(Most of these were cleared by the author response, hence my updated score, see comment below)

---

> ### Author Response · Authors · 2025-06-01
>
> We sincerely thank the reviewer for the constructive feedback, which has strengthened our work. Please find our point-by-point responses below:
>
>
> - **The evaluation is only on two small variants of a single LLM and just QA task.** Thank you for the valuable comments. In response, we have extended our evaluation to include additional LLMs—specifically, a 32B-scaled model and LLaMA-type models—as well as long-form generation tasks beyond QA. The results show that Search-R1 consistently achieves strong performance across different model architectures and a broader range of task types, demonstrating its generalizability and robustness.
>
>
> *Results on other LLMs*
>
> Qwen2.5-32B
>
> | Method | NQ | TriviaQA | PopQA | HotpotQA | 2wikimultihopQA | Musique | Bamboogle | Avg |
> |--|--|--|--|--|--|--|--|--|
> | Direct Inference | 0.2169 | 0.5576 | 0.1915 | 0.2361 | 0.2661 | 0.0517 | 0.1440 | 0.2377 |
> | CoT | 0.2252 | 0.5584 | 0.1936 | 0.2592 | 0.2954 | 0.0806 | 0.5040 | 0.3023 |
> | IRCoT | 0.3058 | 0.6074 | 0.3382 | 0.3616 | 0.4150 | 0.1812 | 0.5280 | 0.3910 |
> | Search-o | 0.2202 | 0.5364 | 0.1615 | 0.1764 | 0.0366 | 0.0604 | 0.3920 | 0.2262 |
> | RAG | 0.3742 | 0.6177 | 0.4089 | 0.3230 | 0.2446 | 0.0736 | 0.2240 | 0.3237 |
> | SFT | 0.3668 | 0.5186 | 0.1682 | 0.2606 | 0.2728 | 0.0993 | 0.1520 | 0.2626 |
> | **Search-R1** | **0.4922** | **0.6686** | **0.4769** | **0.4524** | **0.4546** | **0.2305** | **0.5565** | **0.4760** |
>
>
> Llama3.2-3B
>
> | Method | NQ | TriviaQA | PopQA | HotpotQA | 2wikimultihopQA | Musique | Bamboogle | Avg |
> |--|--|--|--|--|--|--|--|--|
> | Direct Inference | 0.1391 | 0.3682 | 0.1238 | 0.1218  | 0.1066 | 0.0153 | 0.0640 | 0.1341 |
> | CoT | 0.2462  | 0.4866 | 0.1655 | 0.0510 | 0.0827 | 0.0057 | 0.0240 | 0.1517 |
> | IRCoT | **0.3626** | 0.5655 | **0.4282** | 0.2376 | 0.2359 | 0.0719 | 0.2080 | 0.3014 |
> | Search-o1 | 0.1075 | 0.2034 | 0.0929 | 0.1319 | 0.1168 | 0.0348 | 0.1760 | 0.1233 |
> | RAG | 0.3172 | 0.5510 | 0.3371 | 0.2339 | 0.1179 | 0.0343 | 0.0640 | 0.2365 |
> | SFT | 0.3197 | 0.3411 | 0.1220 | 0.2062 | **0.2571** | 0.0641 | 0.1200 | 0.2043 |
> | **Search-R1** | 0.3567 | **0.5776** | 0.3778 | **0.3143** | 0.2330 | **0.0902** | **0.3065** | **0.3223** |
>
>
>
> *Long-form generation task*
>
> | Method | Qwen2.5-3b (ASQA) | Qwen2.5-3b (ELI5) | Qwen2.5-3b (Avg) | Qwen2.5-7b (ASQA) | Qwen2.5-7b (ELI5) | Qwen2.5-7b (Avg) | Qwen2.5-14b (ASQA) | Qwen2.5-14b (ELI5) | Qwen2.5-14b (Avg) |
> |--|--|--|--|--|--|--|--|--|--|
> | direct | 0.2513 | 0.1988 | 0.2250 | 0.3033 | 0.2012 | 0.2523 | 0.2889 | 0.1990 | 0.2439 |
> | RAG | 0.3011 | 0.1927 | 0.2469 | 0.3170 | 0.2018 | 0.2594 | 0.2848 | 0.1932 | 0.2390 |
> | R1 | 0.4244 | **0.2747** | 0.3495 | 0.4367 | **0.2795** | 0.3581 | 0.4442 | **0.2784** | 0.3613 |
> | Search-R1 (PPO) | 0.4801 | 0.2607 | 0.3704 | 0.4709 | 0.2558 | 0.3633 | 0.4421 | 0.2601 | 0.3511 |
> | Search-R1 (GRPO) | **0.4920** | 0.2716 | **0.3818** | **0.5043** | 0.2747 | **0.3895**       | **0.5008** | 0.2729 | **0.3868** |
>
>
> - **No results to show that training is stable.** We conducted three times-repeated experiments and observed that the training reward curves are highly consistent, exhibiting only minor variance. Additionally, the standard deviation of the final performance on the test set is only 0.008, indicating the stability and reproducibility of our training process.
>
> - **Comparison between base and instruct.** Our key observations are as follows: (1) Instruct-tuned LLMs exhibit stronger initial performance and converge more quickly during training; (2) Both base and instruct models ultimately reach similar training reward levels. However, due to the lack of transparency regarding the pretraining and supervised fine-tuning (SFT) data used for Qwen2.5 3B and 7B, it is difficult to make definitive conclusions about their final performance differences. If the SFT data includes reasoning or tool-calling demonstrations, this would naturally give instruct models an advantage in reinforcement learning, potentially leading to better final performance compared to base models. We will clarify this in the revised manuscript.
>
> - **The name of R1.** We will modify the name to “R1-style RL” according to your suggestion.
>
> - **Algorithm 1 (force decoding).** This issue typically occurs when the LLM fails to properly enclose a query or final answer within the expected special tokens. To mitigate this, we insert a prompt that subtly suggests the previous generation may be incorrect, encouraging the LLM to engage in self-reflection and revise its output accordingly.
>
> - **Minor comments in figure 1, page 4, sec 3.3, page 6, tab 2, sec 5.4.** We will modify it according to your suggestions.
>
> - **What is “valid search”?** A valid search is counted when the search engine is successfully called with LLM outputted special tokens.

---

> > ### Comment · Reviewer_udKF · 2025-06-06
> >
> > Thank you for your extensive response and for adding results with additional LLMs and tasks. Assuming these changes get incorporated to the final paper, I've updated my score to 7.

---

> > > ### Author Response · Authors · 2025-06-06
> > >
> > > Thank you very much for your thoughtful follow-up and for updating your score. We're glad the additional results and clarifications addressed your concerns. We will make sure that all the discussed changes are properly incorporated into the final version of the paper.

---

### Official Review · Reviewer_atAs · 2025-05-13

**Rating:** 9
**Confidence:** 5
**Ethics Flag:** 1

**Summary:**

The paper presents a framework for including RAG (particularly search using search engines) during RL phase for LLM training. The paper, titled, Search-R1 discuss the details of this and the challenges overcome. In this framework the authors generate (multiple) search queries during step-by-step reasoning with real-time retrieval and generate rewards for them for RL training. It models the search engine as part of the environment and applies retrieved-token masking to stabilize PPO/GRPO training.

The paper uses Qwen2.5 model family to run their experiments. On seven QA datasets (mix of in-domain and out-of-domain) the authors show that Search-R1 improves performance by 41% (Qwen2.5-7B) and 20% (Qwen2.5-3B) over various RAG baselines under the same setting.

The paper also offers experimental insights into RL optimization methods, LLM choices, and response length dynamics in retrieval-augmented reasoning.

**Reasons To Accept:**

I think this paper is a pretty strong paper. Being able to model the search environment during RL has been of big importance for all practical LLM systems.

1/ This approach gets strong empirical gains -- they achieves 20%-41% EM improvement over RAG baselines

2/ They introduce new tokens to structure the response. The generations are structured with <think>…</think>, <search>…</search>, <information>…</information>, and <answer>…</answer> tokens for clear multi-turn reasoning

3/ They add stability to the RL integration by retrieved-token masking. This boosts performance (e.g., EM jumps from 0.343 to 0.431 on 7B)

4/ This approach seems to generalize across instruction-tuned and non-instruction-tuned models

5/ The paper does thorough ablations comparing PPO vs. GRPO, study of response length dynamics, valid-search behavior

**Reasons To Reject:**

1/ Would have been interesting to see any performance gains on bigger models (Llama models?) to see if the gains hold there as well

2/ It would be interesting to see how performance varies based on choice of knowledge base (search vs wikipedia vs knowledge graph)

---

> ### Author Response · Authors · 2025-06-01
>
> We appreciate your insightful feedback, which has significantly strengthened our manuscript. We address each comment below.
>
> - **Performance on bigger models (llama models).** Thank you for your feedback. In response to your request, we have added results on both the Qwen2.5-32B-Base and LLaMA3.2-3B-Base models.
>
>
> Qwen2.5-32B-Base
>
> | Method | NQ | TriviaQA | PopQA | HotpotQA | 2wikimultihopQA | Musique | Bamboogle | Avg |
> |--|--|--|--|--|--|--|--|--|
> | Direct Inference | 0.2169 | 0.5576 | 0.1915 | 0.2361 | 0.2661 | 0.0517 | 0.1440 | 0.2377 |
> | CoT | 0.2252 | 0.5584 | 0.1936 | 0.2592 | 0.2954 | 0.0806 | 0.5040 | 0.3023 |
> | IRCoT | 0.3058 | 0.6074 | 0.3382 | 0.3616 | 0.4150 | 0.1812 | 0.5280 | 0.3910 |
> | Search-o | 0.2202 | 0.5364 | 0.1615 | 0.1764 | 0.0366 | 0.0604 | 0.3920 | 0.2262 |
> | RAG | 0.3742 | 0.6177 | 0.4089 | 0.3230 | 0.2446 | 0.0736 | 0.2240 | 0.3237 |
> | SFT | 0.3668 | 0.5186 | 0.1682 | 0.2606 | 0.2728 | 0.0993 | 0.1520 | 0.2626 |
> | **Search-R1** | **0.4922** | **0.6686** | **0.4769** | **0.4524** | **0.4546** | **0.2305** | **0.5565** | **0.4760** |
>
>
> Llama3.2-3B-Base
>
> | Method | NQ | TriviaQA | PopQA | HotpotQA | 2wikimultihopQA | Musique | Bamboogle | Avg |
> |--|--|--|--|--|--|--|--|--|
> | Direct Inference | 0.1391 | 0.3682 | 0.1238 | 0.1218  | 0.1066 | 0.0153 | 0.0640 | 0.1341 |
> | CoT | 0.2462  | 0.4866 | 0.1655 | 0.0510 | 0.0827 | 0.0057 | 0.0240 | 0.1517 |
> | IRCoT | **0.3626** | 0.5655 | **0.4282** | 0.2376 | 0.2359 | 0.0719 | 0.2080 | 0.3014 |
> | Search-o1 | 0.1075 | 0.2034 | 0.0929 | 0.1319 | 0.1168 | 0.0348 | 0.1760 | 0.1233 |
> | RAG | 0.3172 | 0.5510 | 0.3371 | 0.2339 | 0.1179 | 0.0343 | 0.0640 | 0.2365 |
> | SFT | 0.3197 | 0.3411 | 0.1220 | 0.2062 | **0.2571** | 0.0641 | 0.1200 | 0.2043 |
> | **Search-R1** | 0.3567 | **0.5776** | 0.3778 | **0.3143** | 0.2330 | **0.0902** | **0.3065** | **0.3223** |
>
>
> The results demonstrate that Search-R1 consistently outperforms strong baseline methods across various model sizes (3B, 7B, 32B) and architectures (Qwen2.5 and LLaMA3.2), highlighting its robustness and generalizability.
>
> - **How performance varies based on the choice of knowledge base or search engine.** This is a great point. We conduct additional experiments to study how the choice of search engine (type of retriever + knowledge source) impacts both the training and inference performance of Search-R1.
>
> *Impact during training.* We evaluate four search engines: (a) Random noise, (b) BM25 + Wikipedia, (c) E5 (ANN) + Wikipedia, and (d) E5 (Exact match) + Wikipedia. Results are shown in the table below. Key findings include: (1) Training with stronger retrievers (e.g., E5 (Exact) and E5 (ANN)) leads to more stable reinforcement learning and better final performance. (2) In contrast, using weaker retrievers (e.g., Random and BM25) significantly limits the final model performance.
>
> | Search Engine | NQ | TriviaQA | PopQA | HotpotQA | 2wikimultihopQA | Musique | Bamboogle | Avg |
> |--|--|--|--|--|--|--|--|--|
> | random | 0.2366 | 0.4941 | 0.1771 | 0.2170 | 0.2688 | 0.0583 | 0.2339 | 0.2408 |
> | BM25 | 0.3413 | 0.6070   | 0.3217 | 0.4043 | 0.3703 | 0.1370 | 0.2800 | 0.3517 |
> | E5 (ANN) | 0.4676 | 0.6214 | 0.3662 | 0.3723 | 0.2872 | 0.1374 | 0.4000 | 0.3789 |
> | **E5 (Exact)** | **0.4806** | **0.6379** | **0.4571** | **0.4328** | **0.3820** | **0.1957** | **0.4240** | **0.4300** |
>
>
> *Impact during inference.* We further evaluate how models trained with different search engines generalize across various inference-time retrievers. The results (table below) show: (1) Search-R1 exhibits strong generalization across retrievers: even when trained with a specific search engine, it performs reasonably well when tested with others. (2) More powerful retrievers at inference time—particularly Google Search (via API)—consistently yield the best results, underscoring the importance of high-quality retrieval in downstream tasks.
>
> | Train / Test Retriever | bm25 | e5 ANN | e5 flat | Google Search |
> |--|--|--|--|--|
> | random | 0.0317 | 0.0317 | 0.0317 | 0.0317 |
> | BM25 | 0.2434 | 0.1587 | 0.2593 | 0.5397 |
> | E5 ANN | 0.2698 | 0.1693 | 0.2540 | 0.6032 |
> | E5 Exact | 0.2487 | 0.1958 | 0.2646 | 0.6032 |
> | **avg** | 0.2540 | 0.1746 | 0.2593 | 0.5820 |

---

> > ### Comment · Reviewer_atAs · 2025-06-09
> >
> > Thanks for the clarifications for my questions.

---

> > > ### Author Response · Authors · 2025-06-10
> > >
> > > Thank you for your thoughtful review and support of our work! We appreciate your engagement with our work.

---

### Decision · Program_Chairs · 2025-07-08

**Decision:**

Accept

**Comment:**

The author study applying PPO and GRPO to a fairly standard multi-hop LLM system. Overall, the authors find that their approach is successful at teaching small Qwen models to improve, learning entirely (as far as I can tell) from final-answer rewards.

There's a very long history for learning from final-answer rewards for these types of system via rejection sampling, with no obvious first such work. However, this was raised and addressed during the review discussion period. This is a CRUCIAL baseline that has to be added in the paper, since every other baseline is an unadapted system that does not even get to learn from any feedback! Other such types of systems include learning from comparisons among rollouts (LeReT; Hsu et al, 2024) or learning from prompt optimization or self-bootstrapped finetuning (which is conceptually akin here to the rejection sampling baseline) of such agents/systems. There's also a long line of work on tuning these systems in more elaborate ways like MDR, IRRR, Baleen, etc. that are simply absent in the paper's discussions.

Overall, and given the unanimous vote to accept by the reviewers, I also recommend acceptance, assuming the authors will surface the baseline (rejection sampling finetuning, or filtered behavior cloning) and the related work discussions in their camera ready.